# Image restoration of degraded time-lapse microscopy data mediated by near-infrared imaging

Nicola Gritti ®[1,2,7], Rory M. Power ®[1,3,7], Alyssa Graves[1] & Jan Huisken ®[1,4,5,6] ✉

Time-lapse fluorescence microscopy is key to unraveling biological development and function; however, living systems, by their nature, permit only limited interrogation and contain untapped information that can only be captured by more invasive methods. Deep-tissue live imaging presents a particular challenge owing to the spectral range of live-cell imaging probes/fluorescent proteins, which offer only modest optical penetration into scattering tissues. Herein, we employ convolutional neural networks to augment live-imaging data with deep-tissue images taken on fixed samples. We demonstrate that convolutional neural networks may be used to restore deep-tissue contrast in GFP-based time-lapse imaging using paired final-state datasets acquired using near-infrared dyes, an approach termed InfraRed-mediated Image Restoration (IR²). Notably, the networks are remarkably robust over a wide range of developmental times. We employ IR² to enhance the information content of green fluorescent protein time-lapse images of zebrafish and *Drosophila* embryo/larval development and demonstrate its quantitative potential in increasing the fidelity of cell tracking/lineaging in developing pescoids. Thus, IR² is poised to extend live imaging to depths otherwise inaccessible.

Time-lapse imaging provides a uniquely dynamic view of biological processes in living systems[1–7]. Powerful insights into development and function have followed through a union of modern microscopes and genetically encoded fluorescent proteins[8,9]. Nevertheless, this approach has its limits in terms of the type of information that can be extracted. For example, the ability to resolve deeply situated tissues in living animals or three-dimensional (3D) cultures is circumscribed by the poor penetration of visible light therein, a challenge exacerbated by the need to maintain physiological conditions. However, in pursuit of a richer biological understanding, we should maximally leverage each specimen to extract complementary information that

is so often left on the table as living samples are typically discarded following a time-lapse. For example, expended samples could exploit the less constrained toolbox available to fixed tissue imaging (multiplexed staining[10]/clearing[11]/expansion[12] or even physical sectioning[13]) or harsh imaging modalities that are less compatible with live imaging but may provide additional information of the specimens final state, such as those that use high illumination intensities[14], long recording times[15], harmful radiation[16] or restrictive mounting[17]. Captured in situ (in a single instrument), this approach can yield multimodal datasets that are useful unto themselves[18]. An intriguing question is whether this additional information extracted from the final state, which is

¹Morgridge Institute for Research, Madison, WI, USA. ²Mesoscopic Imaging Facility, European Molecular Biology Laboratory Barcelona, Barcelona, Spain. ³EMBL Imaging Center, European Molecular Biology Laboratory Heidelberg, Heidelberg, Germany. ⁴Department of Integrative Biology, University of Wisconsin Madison, Madison, WI, USA. ⁵Department of Biology and Psychology, Georg-August-University Göttingen, Göttingen, Germany. ⁶Cluster of Excellence 'Multiscale Bioimaging: from Molecular Machines to Networks of Excitable Cells' (MBExC), University of Göttingen, Göttingen, Germany. ⁷These authors contributed equally: Nicola Gritti, Rory M. Power. ✉e-mail: jan.huisken@uni-goettingen.de

inaccessible to live imaging, can be leveraged to directly enhance the dynamic time-lapse data, ideally, merging the high spatial resolution data obtained at the final time point with the temporal information gained during the time-lapse. Such an approach has been out of reach in the absence of a translation layer between the time-lapse and final-state data; however, supervised machine-learning approaches and in particular, deep neural networks are capable of learning complex, highly nonlinear relationships between two associated datasets[19] and have been applied to a range of bioimage restoration[20–22], segmentation[23–25] and classification tasks[7,26]. Consequently, multimodal microscopy and convolutional neural networks may be used to enhance in vivo time-lapse microscopy images. The deep-learning network could even be trained for cases featuring a single dataset characteristic of snapshots of the live and fixed states and subsequently applied to dynamic live-imaging data.

As an illustrative use case, we consider the origin of image quality degradation deep in tissue. The poor penetration of visible light limits high-resolution fluorescent protein-based imaging to superficial regions in all but the smallest, most transparent embryos or isolated cells. Conversely, near-infrared (NIR; 750–1,750 nm, comprising NIR-I 750–1,000 nm and NIR-II 1,000–1,750 nm) light maintains its directional propagation deeper into tissue[27], as leveraged by two/three-photon microscopy, which relies on the absorption of multiple NIR photons to excite fluorophores with emission spectra in the visible range. Multiphoton microscopy[28] provides sufficient depth penetration for in toto imaging of small animal models such as embryonic/larval zebrafish[29] and Drosophila[30]; however, while the energy deposited and temperature changes induced by the intensely pulsed light have been shown to be safe for imaging small subvolumes of the brains of adult zebrafish[31] and mice[32,33], phototoxic effects take hold long before physical damage is noticeable[34,35] and for in toto imaging of delicate developing embryos and larva, the deleterious influence of multiphoton imaging is often apparent despite efforts to reduce photodamage[30]. Furthermore, serially point-scanned schemes are typically too slow to capture developmental processes. Nevertheless, multiphoton techniques remain a powerful tool in the light microscopy arsenal for intravital imaging and remain the gold standard for deep-tissue fluorescence imaging.

For deep-tissue imaging in developing embryos it is desirable to employ techniques that benefit from the penetration at NIR wavelengths, coupled with the speed and low-intensity requirements of camera-based widefield techniques; however, single-photon NIR schemes are limited by a comparative paucity of live-imaging compatible fluorophores. Although dyes such as indocyanine green are US Food and Drug Administration-approved for use in humans, they[36,37] and other large-molecules[38,39], macromolecular[40] or nanoparticle dyes[41] are not cell-permeable. The imaging of developmental dynamics in small embryos, however, requires that subcellular components or populations of cells can be induced to express fluorescent proteins or selectively labeled. Although improved NIR fluorescent proteins are currently being developed, the most established of these tools only extend partially into the NIR-I with their emission spectra, require visible excitation and suffer from being dim, weakly photostable, often dimeric and require exogenous biliverdin as a chromophoric cofactor[42]. The self-labeling Halo- and SNAP-tagging systems provide the required selectivity and have been used with red dyes to image developing embryos[43] but are limited for NIR imaging by the cell-impermeability of NIR dyes. Likewise, these highly specialized genetically encoded tools are not widely available in animal models, limiting their applicability.

Herein, we demonstrate in vivo time-lapse microscopy with enhanced information content on the basis of paired live (green fluorescent protein; GFP) and final-state (NIR) datasets augmented by a convolutional network to enhance image quality. This technique, termed IR$^2$, is broadly applicable to a multitude of biological systems requiring only GFP contrast for live imaging and post-fixation staining against GFP with NIR dyes. IR$^2$ could thus provide a route to studies of biological dynamics in deeply located tissues and across later developmental stages than hitherto accessible.

## Results

Restoration of contrast using IR$^2$ requires a 1:1 correspondence between the endogenous GFP contrast (live and fixed) and the NIR staining (fixed). First, the instrumentation must be simultaneously capable of fast and gentle imaging of the live state and subsequent capture of the fixed state across a broad spectral region below and above 750 nm. We therefore developed a custom selective plane illumination microscope (IR-mSPIM; Methods), capable of high-resolution imaging over this wide spectral range (Supplementary Note 1; chromatic performance and calibration of the IR-mSPIM). Although a light-sheet microscope compatible with dyes emitting up to 1,700 nm has been reported[38], absorption from water increases substantially from the visible to NIR-I and from NIR-I to NIR-II. While absorption is already appreciable in the NIR-I, this range is commonly considered an optimal window, where scattering and autofluorescence are strongly suppressed relative to the blue-green, while the increased absorption does not cause major heating of tissue or attenuation of the excitation/emission light. Furthermore, the deep-cooled InGaAs cameras required to image beyond 1,000 nm have unfavorable noise characteristics and a cost per pixel >50× that of widespread silicon technologies. For these reasons and the desire to maintain performance in the visible, the IR-mSPIM achieves visible/NIR-I excitation at 488/640/808 nm and efficient collection at bands centered at 525/697/845 nm, alongside compatibility with moderate-to-high NA water-dipping optics, suited to high-resolution live imaging.

Second, staining must be achieved with high specificity and penetration into tissue without harsh treatments that would compromise/deform structures from the organismal level down to the resolution limit of the microscope. Notably, this latter point precludes the use of clearing protocols, which non-uniformly shrink or expand tissues[11]. Nevertheless, due to the diverse compositions of the different biological tissues and organisms, we are unaware of any staining protocol that preserves organismal structures and provides a homogeneous labeling for all tissue types. Rather, protocols need to be finely tuned specifically to each organism and transgenic line (for a description of the protocols used in this work; Methods). This aspect should not be overlooked as, for instance, overfixation of the tissues may decrease the relative brightness of GFP and increase background autofluorescence[44]. Likewise, aggressive permeabilization is precluded by the need to maintain tissue structure/integrity down to the resolution level of the microscope, thus presenting limits to the passage of dye-tagged macromolecules. A discussion of the optimization of the protocols used in this work is provided in Supplementary Note 2.

### Deep-tissue NIR staining and light-sheet imaging

Using IR-mSPIM, we first sought to demonstrate that NIR dye staining against GFP could be achieved with a high degree of selectivity throughout mm-sized embryos/larvae. A fixed transgenic zebrafish larva expressing GFP in the vasculature (Tg(kdrl:GFP)) was imaged after immunostaining with AlexaFluor800 (AF800; Thermo Fisher) (Fig. 1a and Supplementary Table 1). To perform an objective quantification of the quality of infrared (IR) staining, we extracted small volumes (patches of 128 × 128 × 32 pixels each) from the full volumes of visible and IR images, avoiding dark regions of the images, and computed the pixelwise Pearson correlation between the two (Fig. 1b; Methods). As we anticipated better depth penetration in the NIR, we computed the Pearson correlation only for patches within 25 μm from the sample surface. As such, this metric is primarily influenced by the quality of staining as shown under conditions of poor selectivity (Supplementary Fig. 3). Overall, we obtained a Pearson correlation

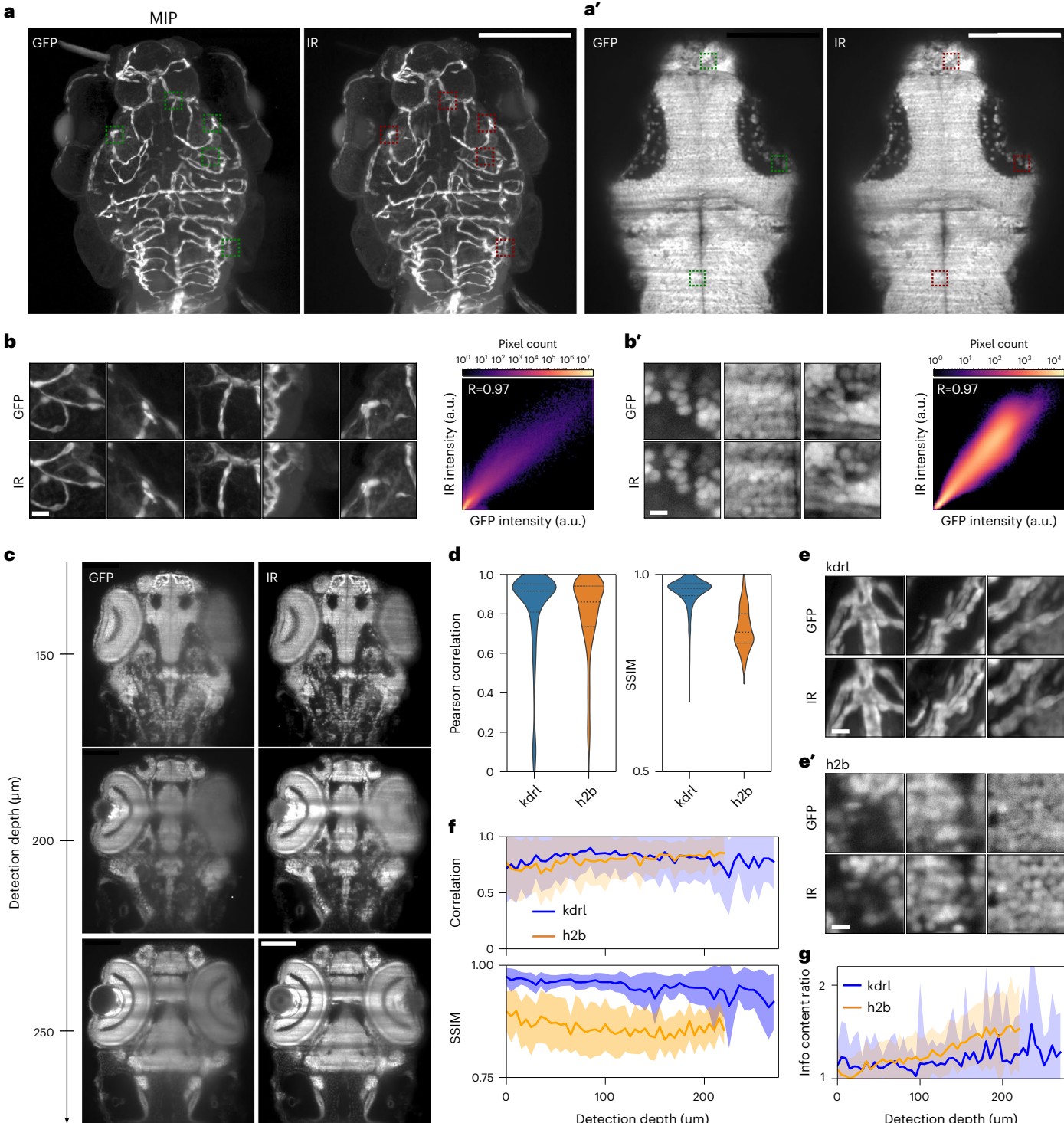

**Fig. 1 | Highly selective near-infrared staining and light-sheet microscopy affords superior imaging at depth in tissue. a**, Maximum intensity projected (MIP) *z* stacks acquired for a fixed Tg(kdrl:GFP) (vascular marker) zebrafish larva (72 hpf) stained against GFP via conventional indirect immunostaining with AlexaFluor800 (AF800). hpf, hours post fertilization. Visible (GFP) left, NIR (AF800) right. Scale bar, 100 µm. **a'**, A single superficial *z* plane from a fixed Tg(h2b:GFP) (nuclear marker) zebrafish larva/embryo (96 hpf) stained against GFP via nanobody-conjugate CF800. Visible (GFP) left, NIR (CF800) right. Scale bar, 100 µm. **b,b'**, Selected superficial patches shown by the dashed boxes in **a,a'**, respectively (visible (GFP) top, NIR (AF800/CF800) bottom) and pixelwise correlation plots for all 125 extracted patches. Scale bar, 5 µm. a.u., arbitrary units. **c**, Multiple deeper *z* planes acquired for the same nuclear marker (h2b) zebrafish embryo/larva shown in **b**. Scale bar, 100 µm. **d**, Pearson correlation and

SSIM for the full *z* stacks acquired for the vascular (kdrl) and nuclear (h2b) marker zebrafish from **a,a'**, respectively. Dashed lines represent 25/50/75th quartiles. **e,e'**, Selected deeper patches for the vascular (kdrl) and nuclear (h2b) marker zebrafish from **a,a'**, respectively. Scale bar, 5 µm. **f**, Pearson correlation and SSIM for all patches extracted at different *z* planes from the full *z* stacks of the vascular (kdrl) and nuclear (h2b) marker zebrafish. The *z* depth provided is the maximum *z* depth into tissue for each image in the stack. The uncertainty envelope is given by the s.d. and the inner thick line represents the mean value. **g**, The information content gain (Methods), between the visible (GFP) and IR channels ($I_{IR}/I_{GFP}$) for all patches extracted at different *z* planes from the full *z* stacks of the vascular (kdrl) and nuclear (h2b) marker zebrafish. The uncertainty envelope is given by the s.d. and the inner thick line represents the mean value.

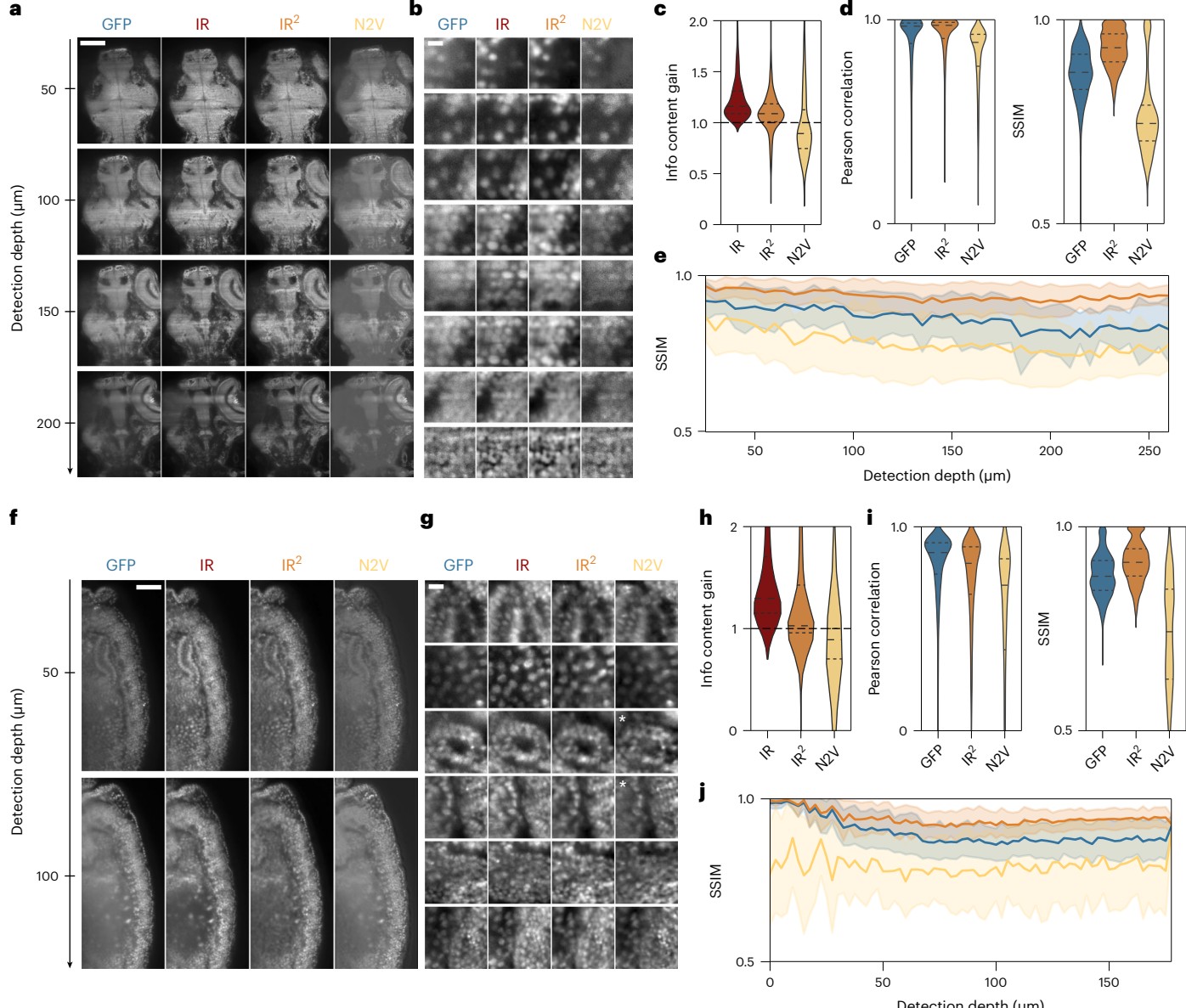

**Fig. 2 | Infrared-mediated image restoration improves image quality of degraded GFP images. a**, Single GFP and IR images (left) extracted at increasing detection depth in a 96 hpf Tg(h2b:GFP) zebrafish larva restored with either IR$^2$ or Noise2Void (N2V) (right). Scale bar, 100 μm. **b**, Example patches for the same zebrafish dataset shown in **a** arranged by increasing cell density. Scale bar, 5 μm. **c**, Violin plots of information content gain (relative to GFP) in patches extracted from the ground-truth (IR, dark red), IR$^2$-reconstructed (IR$^2$, orange) and N2V-reconstructed (N2V, yellow) images. Vertical gray lines indicate s.d. **d**, Pearson correlation and SSIM obtained for patches extracted in the GFP, IR$^2$- and N2V-restored images when compared to the ground-truth image (IR). **e**, SSIM relative to IR image as a function of detection depth for patches extracted throughout the sample. Data are presented as mean ± s.d. **f**, Single z planes at increasing detection depth for a Tg(His2AV-GFP) fly larva (8 hpf) extracted from the input (GFP) and ground-truth image (IR), as well as from restored images obtained from IR$^2$ and Noise2Void. Scale bar, 100 μm. **g**, Example patches for the same fly dataset shown in **f**. White asterisks indicate patches where artifacts were introduced or features were not reconstructed by the Noise2Void network. Scale bar, 5 μm. **h**, Violin plot of information content gain (relative to GFP) in patches extracted from the ground-truth (IR, dark red), IR$^2$-reconstructed (IRIR, orange) and N2V-reconstructed (N2V, yellow) images. **i**, Pearson correlation and SSIM relative to IR image, for GFP, infrared-mediated and Noise2Void reconstructions. **j**, SSIM relative to NIR images as a function of detection depth. Data are presented as mean ± s.d. In all violin plots, dashed lines represent 25/50/75th quartiles.

coefficient close to 1, demonstrating that staining proceeds with high selectivity and without increasing background. The vascular system is highly accessible to large percolating antibodies and so even staining of large samples (mm-sized) can be achieved quickly. We found denser, thicker tissues such as densely packed cell nuclei in the brain, to be more difficult to penetrate. As such we sought alternative staining strategies (Supplementary Note 2; Fixation, permeabilization and staining strategies). Compared to conventional antibodies, nanobodies are substantially smaller and thus potentially better suited

in this regard[45,46]. To explore whether nanobodies could be used to achieve homogenous staining of thick and dense tissues, we conjugated a GFP nanobody (Chromotek) with an NIR cyanine-based fluorescent dye (CF800, Biotium) via maleimide chemistry and stained a zebrafish expressing a histone-GFP fusion Tg(h2b:GFP) (Fig. 1a′ and Supplementary Table 1). Selected patches and Pearson correlation demonstrated comparable selectivity to antibody staining (Fig. 1b′). While traditional immunostaining failed without a more destructive permeabilization (Supplementary Fig. 4), even deeply located tissues showed uniform

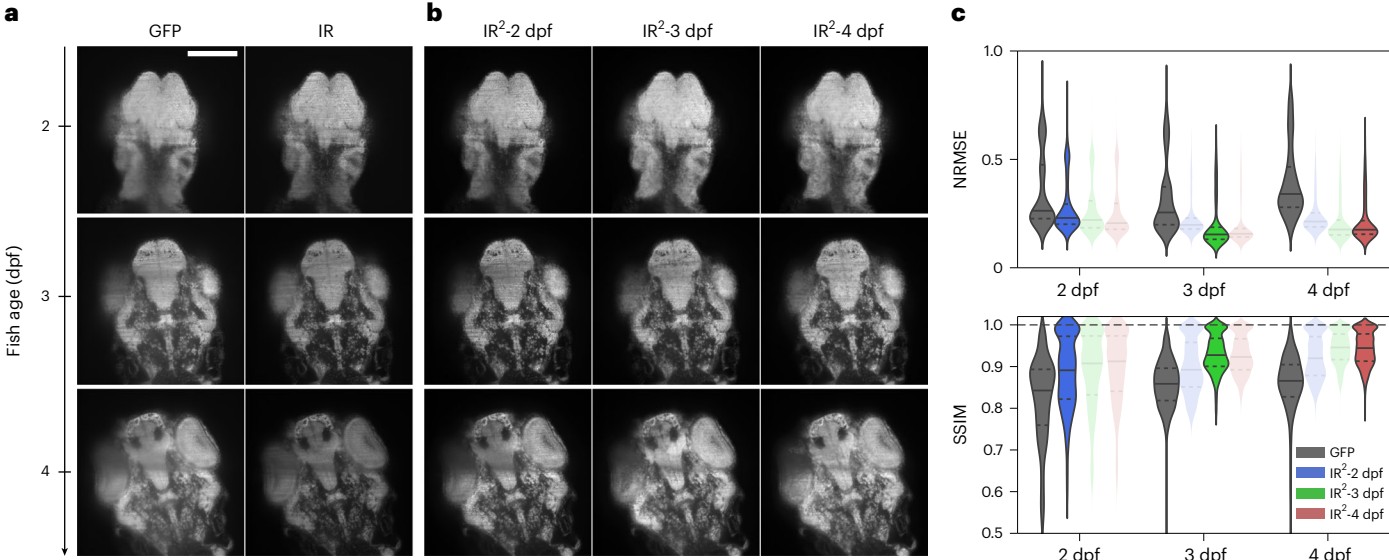

**Fig. 3 | The quality of IR² images is robust over large developmental intervals.** **a**, Representative GFP and IR images of a single *z* plane in a full 3D stack of Tg(kdrl:GFP) zebrafish larvae at 2, 3 and 4 d after fertilization. Scale bar, 100 μm. dpf, days post fertilization. **b**, Image reconstructions obtained from IR² models trained with images from 2, 3 and 4 dpf zebrafish larvae. **c**, Quantification of image quality as measured with the NRMSE and SSIM from the GFP image as well as the images obtained after IR² reconstruction. Values were averaged over all patches extracted from four different samples per age group. Non-transparent violin plots represent the values from the GFP images and the images reconstructed from the IR² network matching the zebrafish age. Transparent violin plots represent reconstruction performed with networks from other developmental stages. Horizontal solid and dashed lines represent the mean and quartile values, respectively.

staining for this challenging case where the dye/nanobody must penetrate through multiple, mildly permeabilized cell membranes and a nuclear envelope (Fig. 1c′).

It is worth noting that the achievable diffraction-limited spatial resolution scales inversely with wavelength; however, the diffraction-limited spatial resolution is really only achieved within a cell layer or two of the surfaces. As such, it is a common practice in light-sheet microscopy to undersample with respect to the Nyquist–Shannon sampling criterion to maximize the field of view. The data of Fig. 1a,b were collected under undersampled conditions for both the GFP and NIR spectral regions. For the denser nuclear data and all data that follow, the spatial sampling was increased to potentially allow finer features to be resolved. In fact, the decrease in spatial resolution in the NIR relative to GFP was less than expected (31/17% in *xy/z*) from a direct comparison of imaging wavelength as described in Supplementary Note 1. The degraded resolution owing to the light-tissue interaction will dominate in any case for the deeply located tissues of interest.

The improvement in image quality in the IR is clearly apparent at depth into tissue and provides a sound basis for image restoration. For both vascular and nuclear transgenic lines, the Pearson correlation clusters strongly toward 1, highlighting that staining is accomplished evenly throughout (Fig. 1d; note the tail toward lower values corresponds to patches dominated by noise (dark regions) of the image and Supplementary Fig. 5). The structural similarity index measure (SSIM; a measure of similarity between two images based on their texture properties[47]; Methods) for the vascular label is also close to 1 (Fig. 1d); the more deeply situated patches demonstrate that due to favorable feature sparsity and size, one can follow individual vessels throughout even for GFP (Fig. 1a,e); however, for the nuclear marker, the majority of patches are clustered around a SSIM of approximately 0.8 (Fig. 1d). In this case, the deeply situated patches are notably different for the GFP and IR channels (Fig. 1e′).

The depth-dependence of the SSIM demonstrates that the deviation between GFP/IR images occurs primarily at depth (Fig. 1f), while the near-depth invariance of the Pearson correlation again highlights its suitability to assess stain penetration. As staining is achieved with a high degree of uniformity, the difference must arise from an improved image quality for the IR channel at depth. To obtain an absolute measure of image quality, we computed entropy-based information content as previously described[48] (Methods), with the IR images at depth showing as much as 2.5× the information content of their GFP counterparts (Fig. 1g). We note that commercially available nanobodies conjugated to the far-red dye AlexaFluor647 (excitation/emission peaks approximately 650/670 nm) performed comparably for staining and offered a more modest improvement to image quality at depth, with the advantage that more common hardware for imaging in the visible spectrum can be used.

## Application of IR² to restore degraded GFP images

Having demonstrated that the IR staining pipeline preserves structure while labeling uniformly throughout tissue depth and selectively for GFP, we considered whether a supervised deep-learning approach[49] could be used to restore a high-contrast image from tissue-scattered GFP images. Image degradation resulting from scattering of a light sheet has been restored using complementary images from two opposed illumination directions[50]; however, this method, proposed also by others[51] remains limited in terms of depth penetration as the associated ground truth arises from comparably superficial regions, whereas tissues that are deeply situated with respect to both illumination directions remain inaccessible. In contrast, we use the superior IR images as a ground truth, thus attempting restoration of images degraded by scattering induced in both illumination and detection. To test whether this approach is generalizable to other model organisms, we used transgenic lines from both zebrafish and *Drosophila* embryos where the cell nuclei are labeled with GFP (Tg(h2b:GFP) and Tg(His2AV-GFP), respectively) and stained them with a GFP nanobody conjugated to the NIR dye CF800. We used these datasets in combination with a common convolutional neural network from the CARE package[20]. First, we implemented an optimized patch extraction routine to minimize the number of dark patches in the training dataset (Supplementary Fig. 6) and to ensure that the IR and GFP patches are maximally aligned locally (Methods). Next, we

used a U-Net deep-learning network[52] and trained it using the GFP and IR images of the nuclear marker as degraded and ground-truth datasets, respectively (Methods). Upon application of the network to the degraded GFP data, restored images emerged with visibly enhanced contrast for both zebrafish and *Drosophila* subjects (Fig. 2a,b,f,g), thus suggesting that the IR² approach could be applied to two model organisms with distinct optical properties.

To benchmark IR² against current restoration methods, we restored both zebrafish and *Drosophila* images using Noise2Void, a self-supervised deep-learning algorithm for image denoising[21] and visually compared the absolute difference map for individual planes and example patches (Methods and Supplementary Fig. 7). To perform a more quantitative analysis, we computed image quality by measuring the gain in information content of IR-, IR²- and N2V-reconstructed images relative to the information content of the input GFP image (Methods show a definition of information content). We observed that the information content gain of N2V images did not outperform that of IR² images, and even showed a lower value for zebrafish samples, suggesting that the degradation of the input GFP images was not dominated by noise (Fig. 2c,h). Instead, we observed that the IR² images from the same samples had significantly higher information content gain, conversely suggesting that scattering at depth was the major source of degradation, and could efficiently be restored using IR² networks (Fig. 2c,h and Supplementary Table 2). The Pearson correlation of IR² images, being primarily sensitive to the staining quality, remained similar across all patches, while we observed a notable decrease for N2V images for both zebrafish and *Drosophila* samples (Fig. 2d,i and Supplementary Table 2). Notably, N2V reconstructions of both samples also displayed a lower SSIM with the ground truth (IR) (Fig. 2d,i and Supplementary Table 2) at all detection depths (Fig. 2e,j), thus suggesting that while N2V networks may learn to efficiently denoise images, they are prone to introduce artifacts (shown for example in Fig. 2g; white asterisks). Additionally, we set out to compare the metrics on a patch-by-patch basis and perform statistical analysis under the null hypothesis that N2V outperforms our IR² approach. This hypothesis was strongly rejected for both zebrafish and *Drosophila* datasets, where we observed that the few images in which N2V outperformed IR² comprised low-information content and were dominated by noise ($P < 0.001$; Supplementary Note 3 and Supplementary Fig. 8 provide details). Overall, both for zebrafish and *Drosophila* samples, we observed a remarkable improvement in the IR²-restored images compared to both the input GFP images and images restored with another deep-learning strategy, which is attributable to a contrast enhancement following restoration.

### Robustness of IR² over long developmental windows

The success of deep-learning-based restorative strategies relies on training data that are representative of the degraded data. When considering the restoration of dynamic time-lapse data from static end point training data, one must consider the efficacy of the restoration over the full time series, over which substantial developmental processes may render morphological changes in the sample. To explore the effect of the developmental interval between the capture of live/degraded and fixed/ground-truth training data, GFP and IR images were captured for nuclear marker zebrafish (h2b:GFP) at 2, 3 and 4 d

after fertilization (Fig. 3a). The GFP/IR from each age group were used to train restoration networks, each of which was subsequently applied to the task of restoring the degraded GFP images for all ages (Fig. 3b and Supplementary Fig. 9). As expected, the network corresponding to the actual age produced the best restoration in all cases between the IR ground truth and GFP degraded data on the basis of normalized root mean square error (NRMSE; Methods) and SSIM (Fig. 3c). Nevertheless, the difference with the restoration based on dissimilar aged training data were small and all restoration networks resulted in an improved similarity with the IR images regardless of age. This suggests that the restoration networks can be successfully applied even over long developmental time windows, while a superior restoration of time-lapse data may be achieved by applying several trained networks over their respective developmental time window. We attribute this robustness to developmental time as a result of the origin of the signal to be restored, in this case, small-scale punctate features arising from cell nuclei largely dominating over large-scale morphological changes.

### Application of IR² for deep-tissue time-lapse imaging

Having demonstrated that IR² performs well over wide developmental periods, we next set out to test whether this approach is applicable to continuous acquisitions of time-lapse live-imaging data. To this end, we performed long-term time-lapse microscopy of developing zebrafish and *Drosophila* embryos whose nuclei are labeled with GFP (Fig. 4a,e), and used IR² networks trained on static images obtained via fixation and nanobody staining of a sample. For both zebrafish and *Drosophila*, we observed a qualitative increase in image contrast throughout the duration of the time-lapse experiment (Fig. 4b,c,f and Supplementary Video 1). To quantify the improvement, we computed the information content gain, defined as the information content of IR² images relative to the information content of the corresponding GFP input for all $z$ slices and time points (Supplementary Fig. 10 and Fig. 4g,h). We then computed the average information content gain in every plane of the $z$ stack and for every time point, thereby obtaining a kymograph representing the spatiotemporal evolution of the quality metric throughout the time-lapse experiment (Fig. 4d,i). In both the zebrafish and the *Drosophila* cases, we observed an improvement in the information content gain with image detection depth. Conversely, the same quantity showed only a minor improvement with time, attributable to a general increase in the sample volume and so a relative improvement for the NIR imaging. This finding is consistent with the observations of Fig. 3, that a single IR² network maintains a similar restoration performance throughout a wide developmental range.

Following demonstration of the image quality improvement achieved by IR² on time-lapse live-imaging data, we next sought to demonstrate how this approach can aid a quantitative analysis common in biological imaging, namely, cell-lineage reconstruction. To this end, we used a new organoid model system, termed pescoids, which consists of zebrafish embryonic explants[53]. Furthermore, to explore the applicability of our approach using more widely available optical and biochemical tools, we used a commercial antibody conjugated with a dye in the far-red (anti-GFP:AlexaFluor647) and a commercially available multiview light-sheet microscope (MuVi-SPIM; Methods). The images showed a clear increase in image contrast at depth (Supplementary Fig. 11). We then trained an IR² network and applied it to

**Fig. 4 | Infrared-mediated image restoration provides high-contrast deep-tissue time-lapse imaging of living biological systems. a**, 3D reconstruction of live Tg(h2b:gfp) zebrafish larva images. Orange boxes represent the regions shown in **b**,**c**. Scale bar, 100 μm. **b**,**c**, Individual $z$ planes at detection depth of 100 μm and 250 μm, respectively for the fish larva shown in **a**. Endogenous GFP (top), IR² reconstructed images (bottom). Scale bar, 100 μm. **d**, Kymograph representing the information content gain relative to the GFP images, for all the images in the time-lapse dataset and as a function of detection depth. Line plots to the right and the bottom represent the depth- and time-average

information content gain. **e**, 3D reconstruction of live (Tg(His2AV-GFP)) fly larva images. Green opaque planes represent the sample sections shown in **f**. Scale bar, 100 μm. **f**, Individual $z$ plane for the fly images shown in **e**. Endogenous GFP (top), IR² reconstructed images (bottom). Red-highlighted time points are shown in **g**,**h**. Scale bar, 100 μm. **g**,**h**, Spatial mapping of the information content gain for the two individual $z$ planes shown. Scale bar, 100 μm. **i**, Kymograph of the information content gain as a function of time and detection depth. Line plots represent the depth- and time-average information content gain.

time-lapse data of pescoids[54] (Methods provides details on mounting and imaging conditions). We observed an increased contrast in the IR² images compared to live GFP data (Fig. 5a and Supplementary Video 2), also quantitatively confirmed by plot profile analysis (Fig. 5b), which persists for all time points (Supplementary Video 3). Next, we used well-established software based on a Gaussian mixture

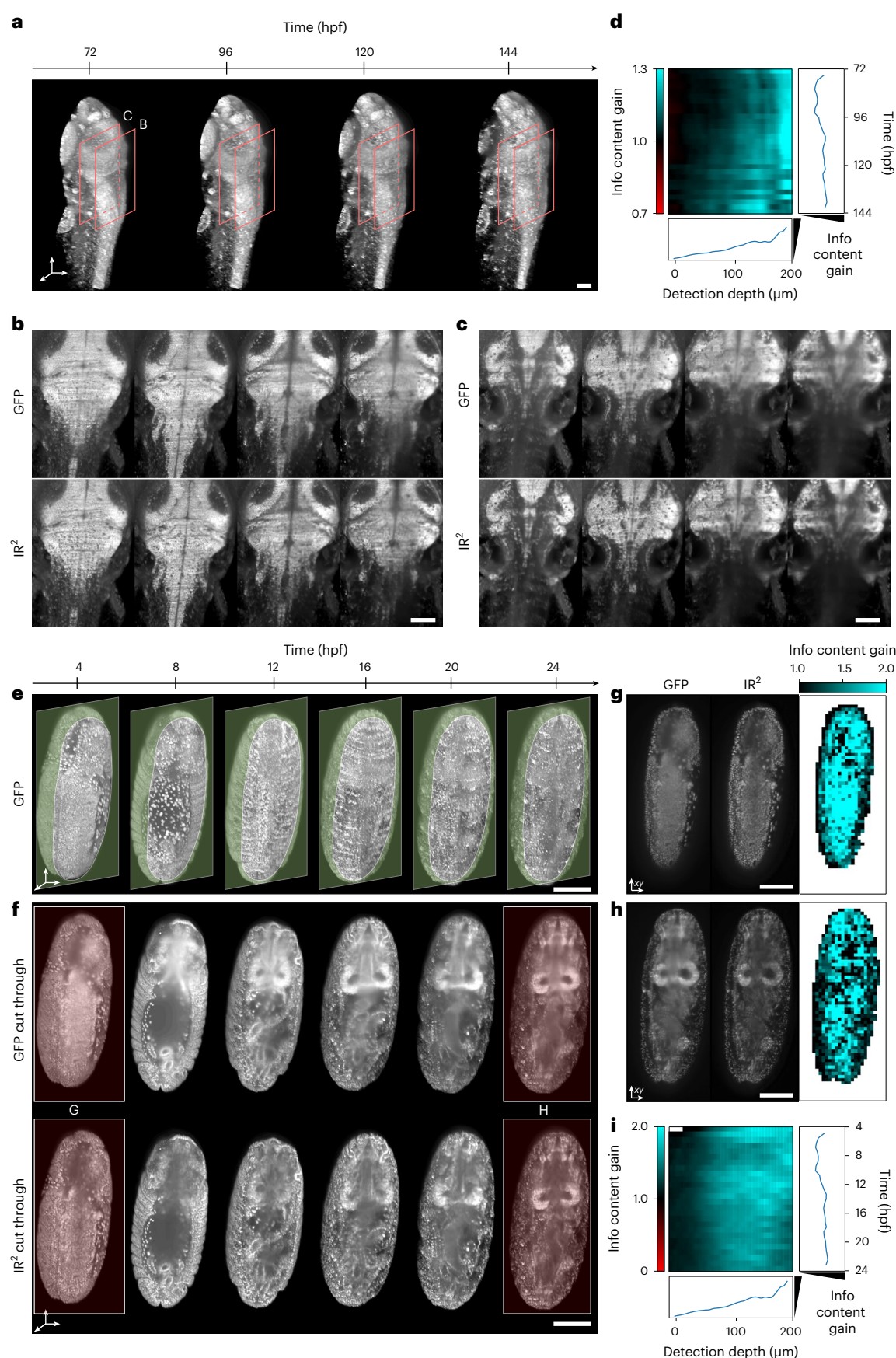

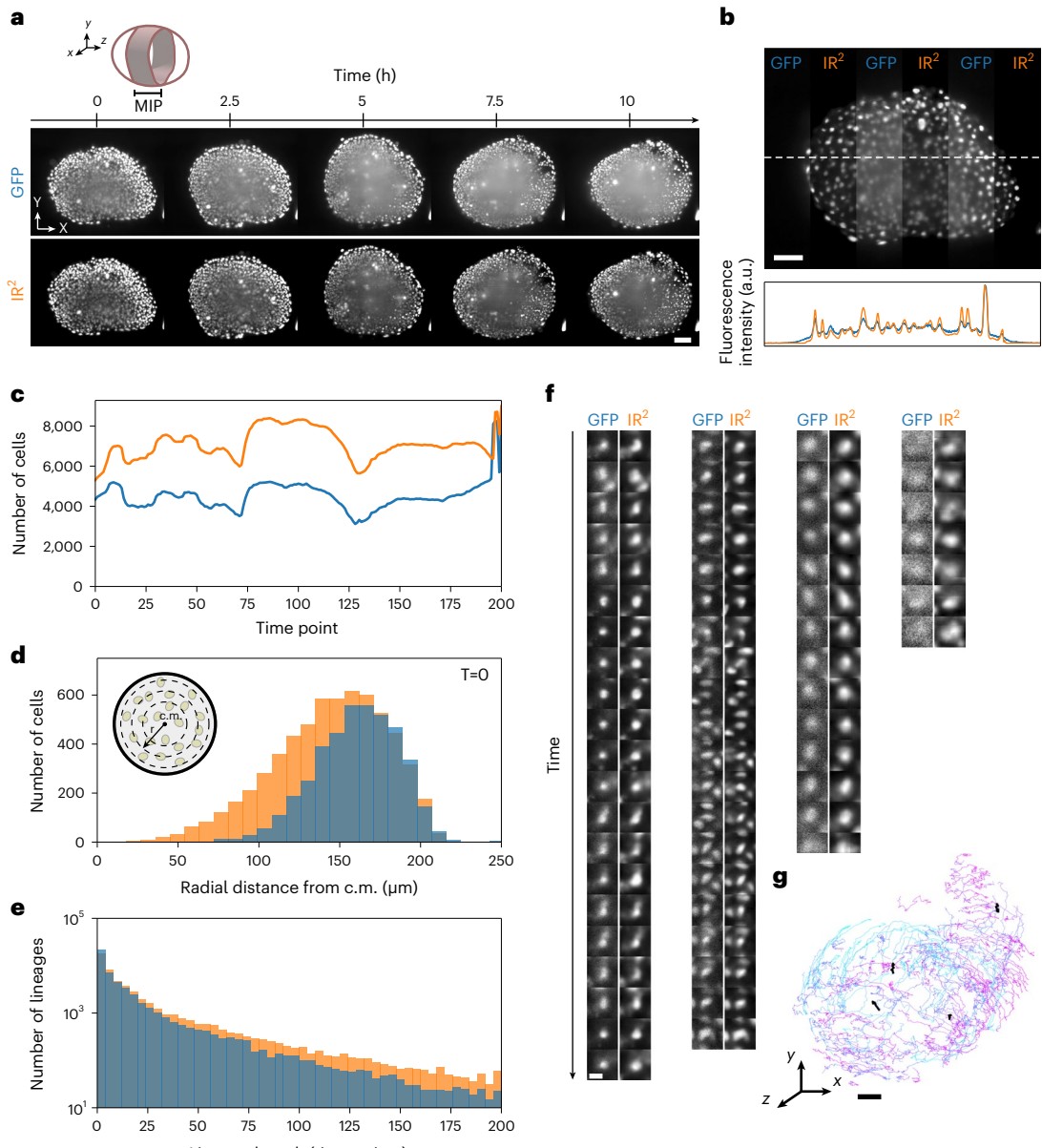

**Fig. 5 | Image restoration of developing pescoid. a**, MIPs of $z$ planes between 240 and 280 μm deep inside a pescoid (cartoon) developing over the course of approximately 10 h. Live GFP images (top) and the images reconstructed using IR² (bottom). Scale bar, 50 μm. **b**, A single $z$ plane from the volume acquired at the first time point, where GFP and IR² images have been alternated in vertical stripes (top). The plot profile along the white dashed line in the GFP (blue) and IR² images (orange) (bottom). Scale bar, 50 μm. **c**, Number of cells detected over time in the GFP (blue) and IR² (orange) volumes. **d**, Number of cells detected in GFP (blue) and IR² (orange) images at the first time point as a function of the distance from the center of mass (c.m.) (inset). **e**, Distribution of track lengths in GFP (blue) and IR² (orange) time-lapse images. **f**, Example of images from cell tracks in GFP and IR² time lapse. Scale bar, 5 μm. **g**, 3D representation of tracks longer than 50 time points, color coded for time (light blue indicates early time points and violet indicates late time points). Black lines indicate tracks represented in **f**. Scale bar, 50 μm.

model to perform cell tracking on both GFP and IR² time-lapse datasets (TGMM)[55]. We observed that TGMM was able to detect a substantially larger number of cells in the IR² images compared to the GFP data at all time points, suggesting that the increased contrast obtained in the reconstructed data greatly aids cell detection (Fig. 5c). Furthermore, as IR² images are expected to provide substantial image quality gain deeper in the sample tissues, we also observed that TGMM was capable of detecting cells in the IR² images in regions where the GFP dataset showed only few detected cells (Fig. 5d). As a consequence of the improved cell detection at all time points, we observed a clear benefit in whole-lineage reconstruction in IR² images, where we obtained substantially longer tracks (Fig. 5e). Evidently, the higher

fidelity on track reconstruction from IR² images is a direct benefit of increased image quality, as observed when visualizing a random subset of all the cell tracks obtained from the reconstructed data in both datasets (Fig. 5f,g).

Taken together, our results demonstrate that the information content of degraded time-lapse microscopy datasets containing only a visible contrast can be augmented by training a neural network on the basis of the relative benefits of deep-tissue NIR imaging and applying it to the task of image restoration to allow time-lapse imaging with high contrast even deep into tissue. Notably, the augmented datasets constitute the basis for better and more accurate quantitative analysis such as for cell-lineage reconstruction.

## Discussion

We have demonstrated that supervised deep learning can be used to restore image quality in deep-tissue images given suitable ground-truth and degraded images. As a first illustration of this concept, we introduce IR$^2$, which exploits NIR dye labeling of GFP as a route to paired degraded and ground-truth datasets, alongside light-sheet microscopy to provide fast and gentle live imaging. Unlike others who have used convolutional neural networks to restore image quality on the basis of reduced scattering at longer wavelengths[56], IR$^2$ offers the following advantages. First, imaging is performed using well-corrected immersion optics with a 3D PSF that is as much as 250× smaller by volume based on the measured resolution in either case[38]. While much of this resolution scaling can be understood by comparison of the numerical apertures and imaging wavelengths employed, we note that achieving resolution congruent with the higher NA of our study, requires careful consideration of aberrations. Nevertheless, a direct comparison is difficult as previously reported NIR-II light-sheet systems define the resolution as measured in tissue, which introduces a sample dependency. The resolution in tissue for IR$^2$ is explored in Supplementary Note 4. Second, IR$^2$ relies on genetically encoded fluorophores and their cognate antibody/nanobody-tagged NIR dyes thus ensuring molecularly precise correspondence between the ground truth and degraded data. Furthermore, the labeling scheme achieves cell permeability and complete staining in a few hours in fixed tissues and so samples may be stained via simple immersion rather than requiring intravenous injection as a delivery mechanism. Coupled with the selectivity noted, one may image arbitrary tissues and subcellular components rather than being limited to imaging the vasculature through non-selective dispersion of the dye in the bloodstream or to cell-surface receptors where circulating dye may bind. Third, IR$^2$ has been developed specifically for the restoration of time-lapse images and has been shown to be robust across wide developmental windows. From a practical implementation point of view, IR$^2$ relies on deep-learning networks that are easy to utilize using a widely used, well-established deep-learning library (CARE)[20]. Implementations for this and similar deep-learning libraries exist for a variety of image analysis ecosystems, including Fiji[57,58], Python and napari plugins[59,60]. Taken together, these aspects open the NIR/deep-learning toolbox to cell and developmental biologists wishing to push live-imaging deeper into tissue. Conversely, IR$^2$ is more limited in terms of maximum imaging depth owing to the NIR-I versus NIR-II operating range. Nevertheless, for imaging with cellular resolution in mm-sized models, the loss of spatial resolution at longer wavelengths likely dominates in a tradeoff as the optical penetration in the NIR-I is typically sufficient for in toto imaging of small embryos/larva.

In contrast to existing restorative deep-learning pipelines such as Noise2Void, which, while powerful in their own domains, are not designed to enhance depth penetration in optical imaging, IR$^2$ has been developed specifically to restore deep-tissue contrast to live-imaging data from zebrafish and *Drosophila* embryos/larvae as well as zebrafish-derived embryonic organoids. En route, we have demonstrated the utility and robustness of this approach to resolve features of embryonic/larval development across wide developmental time windows and which would otherwise be inaccessible owing to the limited penetration of visible light into tissue.

The methods reported can be generalized, requiring only widespread GFP lines and some optimization of staining protocols. Even in the absence of a specialist microscope capable of visible-IR imaging, we showed that improvements to image quality can be made for commercially available GFP-antibody tagged dyes that are efficiently excitable in the far-red range of the spectrum and an appropriate microscope (Supplementary Figs. 4 and 11). The deep-learning networks themselves require only modest computational resources. In this regard, hardware requirements, software and datasets are provided to aid uptake of these restorative abilities by biologists seeking to perform minimally invasive live deep-tissue imaging (Methods and Code Availability provide details).

A potent direction for the future would be to explicitly incorporate a depth dependent component to the restoration, using the detection depth as an additional channel of the input images. Furthermore, model training has been carried out using only single samples, rather than by combining ensembles. Limited training data are a general challenge to deep-learning methods; however, light-sheet techniques are able to generate vast quantities of data rapidly. As no annotation is required, several datasets could be combined to learn additional features for restoration at the cost of increased training time.

We expect further improvements to the performance of IR$^2$ commensurate with developments in fixation protocols that better maintain tissue structure and GFP fluorescence. Similarly, more photostable and brighter IR dyes, red shifted toward the NIR-II[41,61,62] (alongside commensurate developments in low-noise cameras) may allow even deeper tissue imaging. Furthermore, for widely distributed/shared technologies[63], a library of images and restoration networks could be curated for given transgenic lines and shared with other users thus allowing restoration to be applied as an optional part of their post-processing pipeline.

The IR$^2$ restoration network is not limited to the basis of GFP/IR images as degraded/ground-truth pairs and could prove similarly powerful for ground-truth images arising from the use of adaptive optics, multiphoton excitation or indeed chemical clearing if tissue distortions can be obviated. In pursuit of minimizing animal usage, the scheme outlined provides one route by which the information contained in a single subject may be additionally leveraged rather than lost when discarding samples after time-lapse imaging. We anticipate that IR$^2$ can provide a powerful tool in the biologist's arsenal for deep-tissue live imaging.

## Online content

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

## Methods

### Zebrafish husbandry and transgenic lines

Zebrafish (*Danio rerio*) were handled according to established protocols approved by the University of Wisconsin-Madison Animal Care and Use Committee. Zebrafish adults and larvae were maintained on a 14 h/10 h light–dark cycle at 28 °C. Zebrafish embryos were raised in E3 medium at 28 °C. Transgenic lines Tg(kdrl:GFP) and Tg(H2b:GFP) were outcrossed to a casper background to reduce pigmentation where possible. Phenylthiourea was used for depigmentation otherwise. Individual positive embryos were chosen randomly from a clutch of 100–300 embryos (at a density of ~0.5 fish per ml) or from pooled clutches where necessary.

### Fixation and immunostaining of zebrafish

The standard protocol was employed for fixation/staining of transgenic lines that did not show appreciable limitations to penetration of antibodies. Embryos/larvae were fixed in 1.5% paraformaldehyde in phosphate-buffered saline (PBS) + 0.5% Triton (PBST) for 2 h at 4 °C and then washed overnight in aldehyde block (0.3 M glycine in PBST) at 4 °C. The fixed fish were briefly washed in aldehyde block before being permeabilized in PBST for 4 h at room temperature (RT). Subsequently, fish were washed for 1, 2, 5 and 30 min at RT in PBST and blocked for 2 h RT in 0.05% Tween, 0.3% Triton, 5% normal goat serum, 5 % bovine serum albumin (BSA), 20 mM $MgCl_2$ and PBS. After a brief wash in PBST, fish were incubated consecutively overnight at 4 °C and 2 h at RT in primary and secondary antibodies respectively (diluted 1:500) in PBST + 5% goat serum. Finally, embryos/larvae were washed in PBS until they were ready for imaging. In the case of nanobody staining, after the blocking step, fish were incubated for 2 h at 4 °C (diluted 1:500 or 1:100) and then washed in PBS until they were ready for imaging.

The trypsin protocol was carried out for fixation/staining of transgenic lines for which antibodies failed to penetrate tissue when using the standard protocol, the rationale being that a more aggressive permeabilization with trypsin could aid penetration. Embryos/larvae were fixed and washed overnight following the standard protocol. Next, the fish were permeabilized in 0.25% trypsin in PBS for 5 min on ice, washed briefly in PBST and continued from the blocking step from the standard protocol to completion (protocol modified from elsewhere[64]). The protocol was not effective in enhancing penetration of the antibodies (Supplementary Fig. 4).

### Zebrafish mounting

Live embryos and larvae were first anesthetized in E3 medium (without methylene blue) containing 0.16 mg ml$^{-1}$ tricaine (Sigma) and embedded for imaging in fluorinated propylene ethylene (FEP) tubes (ProLiquid, internal diameter (i.d.) 0.8 mm and outer diameter (o.d.) 1.2 mm) containing 1% low-melting-point agarose/E3 (Sigma). Imaging was carried out at RT and the chamber was filled with reverse osmosis (RO) water for fixed samples and tricaine/E3 medium for live samples.

### *Drosophila* husbandry and transgenic lines

Fly stocks were maintained by the laboratory of J. Wildonger at the University of Wisconsin-Madison according to established protocols approved by the University of Wisconsin-Madison Animal Care and Use Committee. Flies were kept on a 12-h light–dark cycle and transferred to fresh vials with food every 2 d. The Tg(His2AV-GFP) transgenic line was used.

### Fixation and immunostaining of *Drosophila*

Embryos were collected at the desired developmental stage, rinsed in RO water and placed for 90 s in a Petri dish with 100% bleach to weaken the outer shell. A paint brush was used to roll the embryos on the Petri dish surface and remove the shell before rinsing with RO water. Embryos underwent a first fixation of 1:1 of 9% paraformaldehyde in PBS:heptane for 30 min RT. The inner vitelline membrane of

embryos was removed by filling the embryo-containing vial with 55% heptane and 45% methanol and striking the vial against a table surface for 2 min, settling for 2 min and repeating three times. Supernatant and floating non-cracked embryos were removed and an aldehyde block (0.3 M glycine in PBST) was added for an overnight incubation at 4 °C. Next, embryos underwent a second fixation with PBST for 4 h at RT, washed with methanol and then ethanol, and blocked with 0.3% Triton X-100, 3% BSA, 10 mM glycine, 1% goat serum, 1% donkey serum and 2% dimethylsulfoxide in PBS at 4 °C overnight. The embryos were incubated in primary antibody (1:500 dilution) overnight at 4 °C in PBS + 5% goat serum, washed twice and then stored in wash buffer overnight at 4 °C (consisting of 0.1% Triton X-100, 3% BSA, 10 mM glycine in PBS and adjusted with NaOH-HCl to pH 7.2. After a brief wash of PBST, the embryos were subsequently incubated for 2 h at RT in secondary antibody solution (1:500 dilution) in PBS + 5% glycine. When using nanobodies, after the blocking step the embryos were incubated for 2 h at 4 °C (1:500 or 1:100 dilution). After antibody/nanobody incubation, embryos were washed in PBS until ready for imaging (protocol modified from elsewhere[65]).

### *Drosophila* mounting

Live and fixed embryos were embedded for imaging in 2% low-melting-point agarose/PBS in FEP tubes with an i.d. of 0.8 mm and an o.d. of 1.2 mm (ProLiquid). Imaging was carried out at RT and the chamber was filled with RO water for fixed samples and PBS for live samples. A number of embryos were mounted in each tube to identify suitably oriented candidates for imaging (with their body axis approximately aligned along the tube axis).

### Fixation and immunostaining of pescoids

Pescoids were generated and collected as previously described[53]. Briefly, zebrafish embryos were cultured at 28 °C in E3 medium until they reached the 256 cell-stage. Embryo cells were then explanted using an eyelash tool, and immediately transferred in L15 medium (Thermo Fisher, 11415049). For fixation, samples were gently washed twice in PBS, transferred in 4% (*w/v*) paraformaldehyde diluted in PBS and fixed at 4 °C overnight. Next, pescoids were transferred in a glass well and gently washed with PBS (three times, 10 min each) and PBSFT (PBS supplemented with 10% fetal bovine serum and 1% Triton X-100) (three times, 10 min each). Immunostaining was performed by incubating the pescoids overnight at 4 °C in a polyclonal antibody (aGFP:AF647, Thermo Fisher, A-31852, 1:500 dilution in PBSFT). The day after, pescoids were washed three times (10 min each) in PBS before imaging.

### Mounting of pescoids

Fixed pescoids were embedded in FEP tubes in 1% low-melting-point agarose/E3 (Sigma) and mounted on a glass capillary with an i.d. of 0.8 mm and o.d. of 1.2 mm (Luxendo/Bruker). Imaging was performed at RT filling the chamber with E3 medium. Live pescoids were embedded in FEP tubes in E3 medium and imaged at 28 °C.

### IR-mSPIM

Visible and NIR excitation was provided by a Toptica MLE laser engine (SM-fiber-coupled: 405 nm, 488 nm, 561 nm, 640 nm all 50 mW) and Omicron, LightHub-4 laser combiner (free-space, LuxX: 685 nm, 50 mW, 785 nm, 200 mW, 808 nm and 140 mW). The collimated laser outputs were expanded in one dimension using pairs of cylindrical lenses. The visible and NIR lasers were combined via a shortpass dichroic mirror. The light sheets were produced by cylindrical lenses, using a galvo mirror-based (Scanlab Dynaxis 3 S) mSPIM scheme to pivot the individual light sheets for efficient stripe suppression[66]. Ultra-broadband achromatic doublets (400–1,000 nm) were used where possible to relay and deliver the light sheets into a sample chamber with coverslip windows via two opposed water-corrected air immersion illumination objectives (Zeiss, LSFM ×10/0.2)

The emission path was optimized for visible-NIR transmission. A multiphoton objective (Olympus XLPLNS10XSSVMP ×10/0.6, 8 mm WD) provided good transmission and moderate NA with a large field of view. Nevertheless, axial chromatic aberration at <650 nm required correction via automatic refocusing of the lens and immersion chamber using a motorized stage Physik Instrumente, M-111.1DG) (typical change in focal plane ± 10 µm). The light sheet remains at a single z plane throughout refocusing and as such the location of the imaged plane does not change as the objective and chamber are moved. The chromatic calibration procedure and performance are discussed in Supplementary Note 1. A tube lens (400–1,300 nm, Thorlabs TTL200MP, 200 mm focal length) was used to produce an image of the sample at ×11.1 magnification on an sCMOS camera (Andor Zyla 4.2), which provides sufficient sensitivity (quantum efficiency >10%) up to ~950 nm. The magnification of the system could be increased to ×22.2 by exchanging the tube lens with an ultra-broadband achromatic lens (Thorlabs, AC508-400-AB-ML). Fluorescence was spectrally filtered from the excitation using bandpass filters (Chroma ET525/50 m, ET697/60 m, ET845/55 m for GFP, AlexaFluor647 and AlexaFluor800/CF800, respectively) mounted on a motorized filter wheel (Ludl 96A351, MAC6000 controller). The reference spectra of the NIR dyes used (AlexaFluor800/CF800) suggest that a combination of a 785-nm laser line and bandpass centered around 820 nm (for 55 nm full-width at half maximum width) would be optimal; however, both antibody/nanobody conjugation were associated with a strong redshifting of the excitation and emission spectra of the IR dyes and the 808-nm laser line and emission filter centered at 845 nm were found to be optimal. This redshifting has been observed in dyes and their conjugates[67,68] and for our purposes is expected to be beneficial resulting in further decreases in scattering and autofluorescence with a small increase in absorption from water. Samples were mounted in FEP tubes via a custom sample holder. Three translation stages and one rotation stage (Physik Instrumente M-111.1DG, U-651 with C-884, C-867 controllers) were used to orient the sample and acquire z stacks. Hardware control and synchronization were provided by custom LabVIEW software and a USB-6343 Multifunction DAQ device (National Instruments).

## Nanobody conjugation

The anti-GFP VHH/Nanobody (Chomotek) underwent site-directed conjugation with the CF680R maleimide (Biotium). The nanobody at concentration of 100 µM was incubated for 2 h at RT with an equimolar amount of dye. Labeled protein was separated from unlabeled protein by size exclusion chromatography.

## Sample imaging

Imaging of zebrafish and *Drosophila* fixed samples was performed on IR-mSPIM (Methods) using 488 nm (GFP) and 808 nm (CF800 dye) excitation wavelengths. The laser powers were chosen to make sure that CF800 images would have an absolute brightness comparable to the GFP ones. For the zebrafish samples, stacks were generated acquiring a z plane every 5 µm using laser powers of <2.2 mW (GFP) and <3.4 mW (CF800) and exposure times of 100 ms for both. Note, laser powers are given as measured in sample medium downstream of the illumination objective. For the *Drosophila* samples, volume stacks were generated acquiring a z plane every 2.5 µm using laser powers of <1.1 mW (GFP) and <3.4 mW (CF800) and exposure times of 100 ms for both.

Live zebrafish and *Drosophila* samples were imaged using laser powers in the previously stated range for fixed tissue GFP imaging; however, the exposure time was set to 20 ms, which is more typical for light-sheet-based live imaging. The laser powers used are comparable to those of previous studies of unimpeded biological development and function using light-sheet microscopy (Supplementary Note 5).

Imaging of fixed pescoids was performed on a commercial light-sheet system (Luxendo MuVi-SPIM, Olympus ×20/1.0NA detection objective, ×16.7 effective magnification, 0.39 µm per pixel), using 10 mW laser power and 50 ms exposure time for both GFP (488 nm wavelength) and AlexaFluor647 (642 nm wavelength). Imaging of live pescoids was performed on the same microscope using 3.5 mW laser power and 50 ms exposure time. In both cases, stacks were generated acquiring a z plane every 2 µm. Live images acquired by the two opposing camera views were registered and fused using the Image Processor module of the Luxendo software (Luxendo processor software v.3.0).

For all images, 3D reconstructions and videos were performed using the Fiji plugin 3DScript[69].

## Deep learning

Upon acquisition of the GFP images and their IR counterparts, we obtained training samples by generating patches of dimension 128 × 128 × 32 pixels throughout the z stack. Patches were extracting using either homogeneous distribution, using a probability of extraction per pixel equal to:

$$P(i,j,k) = \frac{1}{N}$$

Where $N$ is the total number of pixels in the z stack, or using a selective probability:

$$P(i,j,k) = \frac{B}{N_f}, P(i,j,k) = \frac{(1-B)}{N_b}$$

In this case, a threshold was computed using the Otsu thresholding and pixels were classified as foreground (pixel value higher than threshold) or background (pixel value lower than threshold). $N_f$ and $N_b$ represent the total number of foreground and background pixels, respectively. $B$ is a tunable parameter used to adjust the fraction of patches extracted in foreground regions, where a value of $B = N_f/N = (N-N_b)/N$ corresponds to the homogeneous probability distribution. Throughout the experiments, we used $B = 0.9$, thus including only 10% of background patches in the extracted training dataset. Sample coverage was iteratively monitored by comparing the number of foreground and background pixels extracted in the training set, and patch extraction was interrupted when sample coverage reached a value of 95% (Supplementary Fig. 6).

To avoid a misalignment between input and ground-truth datasets due to residual chromatic aberrations, we subsequently performed a correlation-based registration using local translations of the patches and found the (d$x$, d$y$, d$z$) translation that maximized the functional:

$$C(dx, dy, dz) = R(I_i(x, y, z), GT_i(x + dx, y + dy, z + dz))$$

Where $R$ represents the image cross-correlation function, and $I_i$, $GT_i$ represent the input and ground-truth patch, respectively.

With the training dataset thus obtained, we subsequently trained a deep-learning network using the CARE framework[20]. In particular, throughout all experiments, we used a U-Net algorithm with one-channel input and one-channel output, two hidden layers and softmax output layer (Supplementary Fig. 6). The weights of the network were iteratively updated at every epoch using the mean squared error computed between the output of the network and the IR ground truth as a loss function. The input GFP image is thereby transformed at every subsequent layer into a new image with decreased spatial dimensions and increased channel dimension. Patches were divided into training and validation datasets with a ratio of 9:1 and the networks were trained over 100 epochs using a batch size of 8. Depending on the number of patches, training lasted approximately 12–24 h using a GPU Quadro P5000 (16 GB memory) on a CentOS system (512 GB RAM). Prediction of new images was performed on the same computational setup.

All subsequent CPU-based image-based analysis, such as image information content, SSI and root mean square error, were parallelized to use the 80 cores available.

## Image quality assessment

Throughout the paper, we have used four main metrics for image quality and comparison: NRMSE, Pearson correlation coefficient, SSI and entropy-based information content.

**Normalized difference map.** The normalized difference maps shown in Supplementary Fig. 7 are obtained by normalizing the GFP, IR, $IR^2$ and N2V with their respective 0.3 and 99.9 percentiles, to obtain images with identical dynamic range. Next, the absolute values of the difference between pairs of images were computed and shown. For visualization, we randomly chose patches in which the IR image showed substantial image quality improvement compared to the GFP image (patches in which the IR image had an information content gain higher than 1.2).

**Normalized root mean squared error.** This metric is defined as:

$$\text{NRMSE}(x,y) = \frac{\sqrt{<(x-y)^2>}}{<y>}$$

Where $x$ and $y$ are the two images to be compared and $<>$ denotes mean values. Specifically, we used the implementation of NRMSE from the Python package scikit-image[70].

**Pearson correlation coefficient.** Pearson correlation between pairs of images was computed on patches of $(128 \times 128 \times 32)$ pixels extracted from the whole 3D images avoiding dark regions.

For the comparison between GFP and antibody-stained IR images in Fig. 1b,b′, we extracted patches within 20 μm from the surface of the sample. The sample mask was computed with a manually set threshold and the edge mask was obtained subtracting a binary erosion of the mask itself.

**Structural similarity index metric.** SSI is a metric used in image analysis to compare the similarity between two images[47]. As opposed to easier-to-implement measures such as NRMSE and Pearson correlation that rely on absolute pixel values, SSIM is primarily influenced by the structures (or textures) within the images. Briefly, the SSIM between two images $x$ and $y$ is the multiplication of the measures of luminance, contrast and structure. Throughout this work, we have used the implementation of SSIM from the popular Python package scikit-image[70].

**Entropy-based information content.** Similar to previous approaches[48,71], we measured image information content by computing the Shannon entropy of the discrete cosine transform (DCT) of the image patch:

$$IIC = -\sum_{i,j} p_{i,j} \cdot \ln\left(p_{i,j}\right)$$

$$p_{i,j} = \frac{F(i,j)}{\sum_{i,j} F(i,j)}$$

$$F(i,j) = \frac{\text{DCT}(I)^2}{N^2}$$

Where $N$ represents the size of the patch and DCT is the discrete cosine transform of the image patch $I$.

Throughout the text, the information content gain of an image 1 relative to an image 2, is defined as the ratio between the image information content values of the two images.

## Reporting summary

Further information on research design is available in the Nature Portfolio Reporting Summary linked to this article.

## Data availability

A sample of the data is available on the Zenodo repository (https://doi.org/10.5281/zenodo.7075414). Full datasets are available upon request.

## Code availability

The code used to train all models and predict new images, as well as the scripts used for image analysis, are deposited on GitHub (github.com/grinic/2023_InfraRed_Image_Restoration.git). A sample dataset is available on Zenodo (https://doi.org/10.5281/zenodo.7075414).

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

## Acknowledgements

We thank all members of the Huisken laboratory for fruitful discussions on this topic. This work was supported by the Human Frontier Science Program fellowships (LT000227/2018-L; N.G.), (LT000321/2015-C; R.M.P.), the Morgridge Institute for Research (N.G., R.M.P., A.G. and J.H.), the Alexander von Humboldt Foundation (Alexander von Humboldt Professorship; J.H.) and the German Research Foundation (Germany's Excellence Strategy EXC 2067/1-390729940; J.H.). We thank members of the laboratory of J. Wildonger for assistance with *Drosophila* husbandry/handling and provision of transgenic animals. We thank the Biomolecular Screening and Protein Technologies Unit (CRG, Barcelona) for assistance with nanobody conjugation. We thank K. Arato and K. Anlaş from the laboratory of V. Trivedi (EMBL Barcelona) for assistance with pescoid preparation. We thank the Mesoscopic Imaging Facility at the EMBL Barcelona for support with imaging of pescoids.

## Author contributions

N.G., R.M.P. and J.H. conceived of and designed the experiments and analyses. N.G. wrote the image restoration and analysis code. R.M.P. designed and constructed the microscope system. N.G., R.M.P. and A.G. prepared all samples for imaging. N.G., R.M.P. and A.G. performed all imaging. N.G., R.M.P. and J.H. wrote the manuscript. All authors contributed to the discussions and revisions of the manuscript.

## Competing interests

The authors declare no competing interests.

## Additional information

**Correspondence and requests for materials** should be addressed to Jan Huisken.

# Reporting Summary

## Statistics

For all statistical analyses, confirm that the following items are present in the figure legend, table legend, main text, or Methods section.

| n/a | Confirmed | |
|---|---|---|
| ☐ | ☒ | The exact sample size (*n*) for each experimental group/condition, given as a discrete number and unit of measurement |
| ☐ | ☒ | A statement on whether measurements were taken from distinct samples or whether the same sample was measured repeatedly |
| ☐ | ☒ | The statistical test(s) used AND whether they are one- or two-sided<br>*Only common tests should be described solely by name; describe more complex techniques in the Methods section.* |
| ☒ | ☐ | A description of all covariates tested |
| ☒ | ☐ | A description of any assumptions or corrections, such as tests of normality and adjustment for multiple comparisons |
| ☐ | ☒ | A full description of the statistical parameters including central tendency (e.g. means) or other basic estimates (e.g. regression coefficient) AND variation (e.g. standard deviation) or associated estimates of uncertainty (e.g. confidence intervals) |
| ☐ | ☒ | For null hypothesis testing, the test statistic (e.g. *F*, *t*, *r*) with confidence intervals, effect sizes, degrees of freedom and *P* value noted<br>*Give P values as exact values whenever suitable.* |
| ☒ | ☐ | For Bayesian analysis, information on the choice of priors and Markov chain Monte Carlo settings |
| ☒ | ☐ | For hierarchical and complex designs, identification of the appropriate level for tests and full reporting of outcomes |
| ☐ | ☒ | Estimates of effect sizes (e.g. Cohen's *d*, Pearson's *r*), indicating how they were calculated |

*Our web collection on statistics for biologists contains articles on many of the points above.*

## Software and code

Policy information about availability of computer code

| Data collection | For data acquisition, the microscope was controlled in a LabVIEW environment (NI LabVIEW 2017) using custom software, NI multi-function DAQ hardware (NI DAQmx) and various hardware SDKs. Images acquired with Luxendo/Bruker MuVi SPIM microscope were pre-processed with Luxendo software v3.0. |
|---|---|
| Data analysis | The Python code used isavailable at https://github.com/grinic/2023_InfraRed_Image_Restoration.git. A sample dataset is available on the Zenodo repository (doi 10.5281/zenodo.7075414). For training and prediction of deep learning networks, we used the CARE Python3 package with Tensorflow v.2.5.0. Image pro |

For manuscripts utilizing custom algorithms or software that are central to the research but not yet described in published literature, software must be made available to editors and reviewers. We strongly encourage code deposition in a community repository (e.g. GitHub). See the Nature Portfolio guidelines for submitting code & software for further information.

## Data

Policy information about availability of data

All manuscripts must include a data availability statement. This statement should provide the following information, where applicable:
- Accession codes, unique identifiers, or web links for publicly available datasets
- A description of any restrictions on data availability
- For clinical datasets or third party data, please ensure that the statement adheres to our policy

A sample of the dataset used is available on Zenodo (doi 10.5281/zenodo.7075414). Due to the large size of the full dataset (in the order of several TB) and the 50GB limit of Zenodo, we will make the full dataset available upon request.

# Human research participants

Policy information about studies involving human research participants and Sex and Gender in Research.

| | |
|---|---|
| Reporting on sex and gender | No human participants were involved in the study |
| Population characteristics | No human participants were involved in the study |
| Recruitment | No human participants were involved in the study |
| Ethics oversight | No human participants were involved in the study |

Note that full information on the approval of the study protocol must also be provided in the manuscript.

# Field-specific reporting

Please select the one below that is the best fit for your research. If you are not sure, read the appropriate sections before making your selection.

☒ Life sciences  ☐ Behavioural & social sciences  ☐ Ecological, evolutionary & environmental sciences

For a reference copy of the document with all sections, see nature.com/documents/nr-reporting-summary-flat.pdf

# Life sciences study design

All studies must disclose on these points even when the disclosure is negative.

| | |
|---|---|
| Sample size | For each of the model system used, we analyzed 3-5 fixed/stained samples. We did not use a statistical test to calculate sample size,, instead the sample size was empirically determined based on the replicates needed for the deep learning approach. Typically, one sample was used for network training and the other samples for testing and prediction. For the time-lapse microscopy dataset, we used 1 sample per model system as a proof-of-principle. |
| Data exclusions | No data were excluded. |
| Replication | Samples were randomly chosen from a pool of samples laid on the day of the experiments. Data were acquired on multiple days and for different sample batches over the course of several months. All attempts of replication were successful. |
| Randomization | From the pool of zebrafish and drosophila samples, individuals for imaging were chosen randomly. Samples were allocated into experimental group according to their developmental stage. The first acquired sample in each category was generally used to train the deep learning network, and the others were used for testing and prediction. |
| Blinding | Blinding was not relevant to this study due to the randomization used and no categorization of the data. |

# Reporting for specific materials, systems and methods

We require information from authors about some types of materials, experimental systems and methods used in many studies. Here, indicate whether each material, system or method listed is relevant to your study. If you are not sure if a list item applies to your research, read the appropriate section before selecting a response.

## Materials & experimental systems

| n/a | Involved in the study |
|---|---|
| ☐ | ☒ Antibodies |
| ☒ | ☐ Eukaryotic cell lines |
| ☒ | ☐ Palaeontology and archaeology |
| ☐ | ☒ Animals and other organisms |
| ☒ | ☐ Clinical data |
| ☒ | ☐ Dual use research of concern |

## Methods

| n/a | Involved in the study |
|---|---|
| ☒ | ☐ ChIP-seq |
| ☒ | ☐ Flow cytometry |
| ☒ | ☐ MRI-based neuroimaging |

## Antibodies

| | |
|---|---|
| Antibodies used | Primary: anti-GFP (ThermoFisher, A-11122), nano-GFP (Chromotek, GT-250)<br>Secondary: AF800 (ThermoFisher, A-32808), CF800 (Biotium, #92128), AF700 (TermoFisher, A-21038)<br>Conjugated: nano-GFP+AF647 (Chromotek, GB2AF647) |
| Validation | Each antibody was validated by the commercial company providing the product. |

Validation

The primary anti-GFP has been used extensively for immunohistochemistry in various model systems such as drosophila, zebrafish and mouse. Relevant literature can be found at:
https://www.thermofisher.com/antibody/product/GFP-Antibody-Polyclonal/A-11122.

For the nano-GFP (GT-250), relevant literature can be found at:
https://www.ptglab.com/products/GFP-VHH-recombinant-binding-protein-gt.htm#publications,
and Chromotek provides the following statements:
Alpaca anti-GFP VHH, purified recombinant binding protein for extraordinary stable & reliable binding
• GFP-VHH:GFP complex is stable up to 80 °C, 1 mM DTT, 3 M Guanidinium•HCl, 8 M Urea, 2 M NaCl, 2 % Nonidet P40 Substitute, 1 % SDS, 1 % Triton X-100, 3 % Deoxycholate
• Fulfills highest requirements for antibody validation
• Structure and function are characterized

For the conjugated nanobody (GB2AF647), relevant literature can be found at:
https://www.ptglab.com/products/GFP-Booster-Alexa-Fluor-647-gb2AF647.htm
 Chromotek provides the following description:
The GFP-Booster stabilizes, enhances, and reactivates the signal of GFP-fusion proteins. Due to its small size, the GFP-Booster enables higher image quality in epifluorescence, confocal, and super-resolution microscopy:
• Considerably higher tissue penetration rates
• Superior accessibility and labelling of epitopes in crowded cellular/organelle environments
• Less than 2 nm epitope-label displacement minimizes linkage error
• Monovalent VHHs do not cluster their epitopes
• Validation: structure and function characterized
• Consistent and reliable performance due to recombinant production

## Animals and other research organisms

Policy information about studies involving animals; ARRIVE guidelines recommended for reporting animal research, and Sex and Gender in Research

Laboratory animals

Zebrafish: transgenic lines Tg(kdrl:GFP) and Tg(h2b:GFP) at larval stages between 24 and 144 hours post fertilization.
Drosophila: Tg(his2av:GFP) at embryo stages between 4 and 24 hours post fertilization.

Wild animals

The study did not invlove wild animals.

Reporting on sex

The study does not apply to one sex only.

Field-collected samples

The study did not include samples collected from the field.

Ethics oversight

Zebrafish (Danio rerio) were handled according to established protocols approved by the 380 University of Wisconsin-Madison Animal Care and Use Committee.
Fly stocks were maintained by the lab of Jill Wildonger at the University of Wisconsin-Madison 420 according to established protocols approved by the University of Wisconsin- Madison Animal 421 Care and Use Committee.

Note that full information on the approval of the study protocol must also be provided in the manuscript.

