## [Peer Review File · Nature Methods]

Peer Review Information

Manuscript Title: Image restoration of degraded time-lapse microscopy data mediated by near-infrared imaging.

Corresponding author name(s): Jan Huisken

Editorial Notes: None

Reviewer Comments & Decisions:

Decision Letter, initial version:

Dear Jan,

Your Article entitled "Image restoration of degraded time-lapse microscopy data mediated by infrared-imaging." has now been seen by two reviewers, whose comments are attached. While they find your work of potential interest, they have raised serious concerns which in our view are sufficiently important that they preclude publication of the work in Nature Methods, at least in its present form.

As you will see, the reviewers raise concerns about the level of conceptual advance, the way the work is presented in terms of referencing and discussing the literature, the benefits relative to existing tools, and the practical benefits of your methods.

We appreciated that the conceptual advance was limited before we sent the paper for review, so we are not penalizing you for that now. However, in cases like this, where related work has been done, the practical advance needs to be unmistakable to justify publication in Nature Methods. We thus ask you to focus your revision (if you plan to revise) on addressing the referee concerns and both showing experimentally and explaining clearly the benefits of your approach over existing methods, especially for imaging development.

Should further experimental data allow you to fully address these criticisms we would be willing to look at a revised manuscript (unless, of course, something similar has by then been accepted at Nature Methods or appeared elsewhere). This includes submission or publication of a portion of this work

somewhere else. We hope you understand that until we have read the revised paper in its entirety we cannot promise that it will be sent back for peer-review.

If you are interested in revising this manuscript for submission to Nature Methods in the future, please contact me to discuss your appeal before making any revisions. Otherwise, we hope that you find the reviewers' comments helpful when preparing your paper for submission elsewhere.

Sincerely,
Rita

Rita Strack, Ph.D.
Senior Editor
Nature Methods

Reviewers' Comments:

Reviewer #1:

Remarks to the Author:

In this manuscript, the authors address a very important issue of image degradation at depth in whole-mounted specimens. Given the great and growing desire to capture volumetric images of genetically encoded labels, immunocytochemistry, axonal transport of labels, spatial transcriptomics, etc... this is a very timely and important issue. Even the best optics, with the thinnest light sheets, or the most optimized objective lenses, microscopes will be unable to escape the degradation of the image with depth due to light scattering. Scattering will both degrade the image and attenuate the signal. The topic of this paper is important and will grow in importance with time, making it appropriate for the readership of the journal.

As light scattering is strongly dependent on the wavelength of the light used to excite or detect fluorescence, the authors propose an exciting solution, using the double staining of a specimen with a long-wavelength dye to generate a "ground truth" image of the visible light GFP at wavelengths much longer and hence much less scattered. This gives them key data on the specimen's scattering and other image degrading properties so that a neural network can be trained to compensate for the losses/degradations.

This neural network seems surprisingly robust, with corrections processed based on data from one stage of development being able to compensate for image losses and degradations at other stages. This strike me as both good news, and a challenge, as this broad applicability suggests that it should be possible to improve the processing even further, with transfer learning, etc.

There are many aspects of the current version of the manuscript that strongly support the publication of a study of this sort, and there are many aspects that argue that it is in need of major revision. Note that I am very supportive of a revised version of this work being published in a high-visibility journal, and that I agree with many if not most of the claims. However, the figures and the text are not convincing. These issues, which will be simple to resolve, include:

1. Text issues.

a. The references on the importance of time lapse imaging of specimens are arbitrary and amusing at best, and worrisome in many ways. The choices seem scientifically questionable, and while popular and published in high-impact journals, they hardly are the earliest or best examples of the approaches. In some cases, they are important in their conclusions, but not really excellent examples of the best of volumetric imaging over time. The authors should improve their scholarship in motivating this work and putting it in context.

b. Similarly, the importance of three-photon and two-photon imaging seems to be dismissed, which does not build credibility or make the present manuscript more appropriate for publication. The authors are correct that these approaches are wrong for many experimental questions, but the matching of imaging tool/approach with scientific problem is as true for laser scanning as it is for light sheet. The inappropriateness of the techniques in this manuscript for some uses should not make it less attractive to publish.

c. The discussion of IR absorbance by water is partially wrong, as the absorbance at 800nm is more than 100-fold higher than the absorbance in the blue.... Not the negligible difference that seems to be implied by the hugely increased absorbance in the NIR that is discussed. Again, some balance will not weaken the point and increase credibility.

d. Presenting the camera sensitivity as “not-negligible” is an odd phrasing – perhaps amusing but not informative.

e. The statement about the resolution required to see nuclei is incorrect. The diameters of the nuclei are less critical than the spacing between the nuclei, which can be significantly smaller.

f. The comment on the red-shifting of the dye is telegraphic at best.

2. The presentation of the data is not of high quality and there is minimal definition of what has been performed. Given that this is intended as a high-visibility guide on how to perform an important imaging processing, and analyzing the results this is troublesome.

a. For example, on lines 192/3, the authors state “SSIM showed a remarkable improvement in the IR2 restored images, both for zebrafish and Drosophila (mean, SD = XX, XX respectively for GFP and restored mages respectively).... Even with XX, XX filled in, this sentence will not be understandable.

b. SSIM is defined as an abbreviation for structural similarity index, but not as a technique or a citation/explanation of what it does and how to use it, so readers will be neither be able to critically evaluate the claims made by the authors nor be able to reproduce this approach for themselves.

c. Pearson’s correlation seems to have included black pixels in the analysis, which for some image times will give misleading results about the correlations in the features that are observed. I suspect that if the black pixels (or near black pixels) were to be removed as has been performed in many of the best papers, the conclusions would have been significantly different.

d. There should be statistical evaluations of the differences that are offered, with the necessary Bonferroni multiple comparisons corrections.

3. The figures are disappointing and variable.

a. Panels have an odd mix of whisker and violin plots.

b. Figures present Pearson’s analyses with the scaling dominated by the black (0,0) pixels so that patterns, if they are there, cannot be discerned.

c. The comparison of images with depth assumes that a reader can judge differences in the images. A difference map should be offered, with a color scale and an offset that makes it simple for the reader to judge errors of commission and omission.

d. The N2V denoising neural network outperforms the authors’ technique in some ways (by SSIM plot), and without the difference map requested in 3c, the comments from the authors on its poor performance are not supported (the only metrics that are offered show it is superior).

e. The figure of IR-dye (AF700) staining is very bad. If the figure is an accurate rendering, it is a failed experiment and the comparison offered in Supp Fig 1 is meaningless.

f. There are claims made about the axial and lateral chromatic aberrations, which should be shown not just dismissed.

In summary, I am very supportive of this work and urge the authors to make it clear, accessible, bolstered by data that is offered easily, not on request.

Reviewer #2:

Remarks to the Author:

The authors proposed the use of a deep learning method (convolutional neural networks) to restore deep-tissue contrast in GFP-based visible imaging using paired final-state datasets acquired using NIR-I dye. Frankly, a similar idea has been published in 2021 (<https://doi.org/10.1073/pnas.2021446118>), in which they used a deep-learning-based approach to transform a blurred NIR-I (800-1000 nm) image to a much higher-clarity image like in NIR-IIb (1500-1700 nm) to remove the scattering effect in the NIR-I window. On the other hand, the authors used a common convolutional neural network from the published CARE package (Ref.19, Nat. Methods 15, 1090–1097 (2018)). The idea and the methods in this manuscript are not new. So I do not recommend publication except the authors give more evidence to show its novelty.

The following are some questions/comments about this manuscript.

1. The idea of this manuscript is similar to a published article (<https://doi.org/10.1073/pnas.2021446118>) which enhanced NIR-I (800-1000 nm) imaging by NIR-IIb (1500-1700 nm) imaging using deep learning methods. What is the advantage of the authors' methods?
2. The optical diffraction limit confined resolution of IR imaging is lower than visible/GFP imaging. Did the authors consider this influence when they performed IR2 (infrared-mediated image restoration)?
3. The authors used a common convolutional neural network from the published CARE package (Ref.19). What is new?
4. According to the lasers and fluorescence bandpass filters in the Methods, the applied wavelength window is ~ 400-800 nm. However, the authors used "infrared" in the title and all over the manuscript. It is not accurate. "Near-infrared" or "NIR-I" would be better.
5. Line 66, the authors claimed that "but suitable NIR fluorophores are incompatible with live imaging (a discussion of infrared imaging is given in Supplementary Note 1)." This claim is quite confusing as a lot of NIR dyes have been used for live imaging. Some of them such as ICG has been approved by FDA for human imaging.

In Supplementary Note 1, for multiphoton microscopy, "these delicate samples are damaged by the intensely pulsed light and develop too quickly for their dynamic processes to be captured via serially point-scanned schemes." Recently many groups have demonstrated the laser power they used is safe,

such as the work done by Prof. Chris Xu (Nature Methods 15, 789–792 (2018)). Other groups also developed ultrafast two-photon fluorescence microscopy with kilohertz speed (Nature Methods 17, 287–290).

6. The authors claimed, “Although a light sheet microscope compatible with dyes emitting up to 1700 nm has been reported²⁷, absorption from water increases > 100x compared to an optimum window at 800 - 950 nm²⁶ contributing to heating and attenuation of the excitation/emission light”. How the authors did this calculation? Have the authors measured the water absorption by themselves? “100x” is much larger than the reviewer’s measurements.

7. Line 98, “overfixation of the tissues may decrease the relative brightness of GFP and background autofluorescence.” Do the authors mean “increase” background autofluorescence?

8. Line 101, “A discussion of the optimization of the protocols used in this work is provided in Supplementary Note 1.” I did not see the mentioned discussion in Supplementary Note 1.

9. Why the GFP results and IR results in Fig. 1A and 1A’ do not show an apparent difference but the Fig. 1C shows a difference?

10. What is the exposure difference between the GFP images and IR images in Fig. 1E’? It looks like the GFP used a shorter exposure time. What are the exposure time and frame rate of the images in this manuscript? What is the fluence of the laser? Are they safe for small animals? Details are needed.

11. Line 170, “Next, the restoration network was trained (see Supplementary Note/Methods) using the GFP and IR images of the nuclear marker as degraded and ground truth datasets, respectively.” I did not see the restoration network or training methods in the Supplementary Note.

12. “The information content gain of an image 1 relative to an image 2, is defined as the ratio between the information content values of the two images.” What are the “information content values”?

13. In Fig. 2G, I did not observe the obvious difference between GFP, IR and IR2 images.

14. Line 193, “(mean, SD = XX, XX respectively for GFP and restored images respectively),” What is “XX”?

15. For Fig. 3A, do the authors have IR raw images at 2-, 3- and 4-days post fertilization for comparison or used as ground truth?

Author Rebuttal to Initial comments

Reviewer #1:

Remarks to the Author:

In this manuscript, the authors address a very important issue of image degradation at depth in whole-mounted specimens. Given the great and growing desire to capture volumetric images of genetically encoded labels, immunocytochemistry, axonal transport of labels, spatial transcriptomics, etc... this is a very timely and important issue. Even the best optics, with the thinnest light sheets, or the most optimized objective lenses, microscopes will be unable to escape the degradation of the image with depth due to light scattering. Scattering will both degrade the image and attenuate the signal. The topic of this paper is important and will grow in importance with time, making it appropriate for the readership of the journal.

As light scattering is strongly dependent on the wavelength of the light used to excite or detect fluorescence, the authors propose an exciting solution, using the double staining of a specimen with a long-wavelength dye to generate a "ground truth" image of the visible light GFP at wavelengths much longer and hence much less scattered. This gives them key data on the specimen's scattering and other image degrading properties so that a neural network can be trained to compensate for the losses/degradations.

This neural network seems surprisingly robust, with corrections processed based on data from one stage of development being able to compensate for image losses and degradations at other stages. This strikes me as both good news, and a challenge, as this broad applicability suggests that it should be possible to improve the processing even further, with transfer learning, etc.

There are many aspects of the current version of the manuscript that strongly support the publication of a study of this sort, and there are many aspects that argue that it is in need of major revision. Note that I am very supportive of a revised version of this work being published in a high-visibility journal, and that I agree with many if not most

of the claims. However, the figures and the text are not convincing. These issues, which will be simple to resolve, include:

1. Text issues.

a. The references on the importance of time lapse imaging of specimens are arbitrary and amusing at best, and worrisome in many ways. The choices seem scientifically questionable, and while popular and published in high-impact journals, they hardly are the earliest or best examples of the approaches. In some cases, they are important in their conclusions, but not really excellent examples of the best of volumetric imaging over time. The authors should improve their scholarship in motivating this work and putting it in context.

We thank the reviewer for the comment, we have now updated the literature to include the latest and more relevant publications, including:

- Inoue, S. 1953. Polarization optical studies of the mitotic spindle 1. The demonstration of spindle fibers in living cells. *Chromosoma*. 5:487–500.
- Paddock, Steve. 2001. "A Brief History of Time-Lapse." *BioTechniques* 30 (2): 283–89. <https://doi.org/10.2144/01302bt01>.
- Ruffins, Seth W., Russell E. Jacobs, and Scott E. Fraser. 2002. "Towards a Tralfamadorian View of the Embryo: Multidimensional Imaging of Development." *Current Opinion in Neurobiology* 12 (5): 580–86. [https://doi.org/10.1016/s0959-4388\(02\)00366-5](https://doi.org/10.1016/s0959-4388(02)00366-5).
- Megason, Sean G., and Scott E. Fraser. 2003. "Digitizing Life at the Level of the Cell: High-Performance Laser-Scanning Microscopy and Image Analysis for in Toto Imaging of Development." *Mechanisms of Development* 120 (11): 1407–20. <https://doi.org/10.1016/j.mod.2003.07.005>.

- Landecker, Hannah. 2009. "Seeing Things: From Microcinematography to Live Cell Imaging." *Nature Methods* 6 (10): 707–9. <https://doi.org/10.1038/nmeth1009-707>.

b. Similarly, the importance of three-photon and two-photon imaging seems to be dismissed, which does not build credibility or make the present manuscript more appropriate for publication. The authors are correct that these approaches are wrong for many experimental questions, but the matching of imaging tool/approach with scientific problem is as true for laser scanning as it is for light sheet. The inappropriateness of the techniques in this manuscript for some uses should not make it less attractive to publish.

We fully agree with the reviewer on this point. We had tried to highlight how our technique fits into the existing infrared imaging landscape with supplementary note 1 but clearly such an important topic warrants inclusion in the main text and expansion of the discussion. We have added the following passages:

"Conversely, near infrared (NIR, 750 - 1750 nm, comprising NIR-I 750 - 1000 nm, NIR-II 1000 - 1750 nm) light maintains its directional propagation deeper into tissue²⁷, as leveraged by two/three-photon microscopy, which relies on the absorption of multiple NIR photons to excite fluorophores with emission spectra in the visible range. Multiphoton microscopy²⁸ provides sufficient depth penetration for in toto imaging of small animal models such as embryonic/larval zebrafish²⁹ and drosophila³⁰. However, while the energy deposited and temperature changes induced by the intensely pulsed light have been shown to be safe for imaging small sub-volumes of the brains of adult zebrafish³¹ and mouse^{32,33}, phototoxic effects take hold long before physical damage is noticeable^{34,35} and for in toto imaging of delicate developing embryos and larva, the deleterious influence of multiphoton imaging is often apparent despite efforts to reduce photodamage³⁰. Furthermore, serially point-scanned schemes are typically too slow to capture developmental processes. Nevertheless, multiphoton techniques remain a powerful tool in the light microscopy arsenal for intravital imaging and remain the gold-standard for deep tissue fluorescence imaging.

For deep tissue imaging in developing embryos it is desirable to employ techniques which benefit from the penetration at NIR wavelengths, coupled with the speed and low intensity requirements of camera-based widefield techniques. However, single-photon NIR schemes are limited by a comparative paucity of live-imaging compatible fluorophores. Although dyes such as indocyanine green are FDA-approved for use in humans, they^{36,37} and other large-molecule^{38,39}, macromolecular⁴⁰ or nanoparticle dyes⁴¹ are not cell-permeable. The imaging of developmental dynamics in small embryos however, requires that subcellular components or populations of cells can be induced to express fluorescent proteins or be selectively labeled. NIR FPs only extend partially into the NIR-I with their emission spectra, require visible excitation, and suffer from being dim, weakly photostable, often dimeric and require biliverdin as a chromophoric co-factor⁴². The self-labeling Halo- and SNAP- tagging systems provide the required selectivity and have been used with red dyes to image developing embryos⁴³ but are limited for NIR imaging by the cell-impermeability of NIR dyes. Likewise, these highly-specialized genetically-encoded tools are not widely available in animal models, limiting their applicability."

c. The discussion of IR absorbance by water is partially wrong, as the absorbance at 800nm is more than 100-fold higher than the absorbance in the blue.... Not the negligible difference that seems to be implied by the hugely increased absorbance in the NIR that is discussed. Again, some balance will not weaken the point and increase credibility.

We thank the reviewer for noting this point, they are indeed quite correct. We have expanded the discussion of absorption due to water as follows. Please see the main text for literature referencing.

"Although a light sheet microscope compatible with dyes emitting up to 1700 nm has been reported³⁸, absorption from water increases substantially at longer NIR wavelengths. We compiled water absorption data⁴⁴⁻⁴⁶ and define 5 spectral bands each of 55 nm width: 500 - 555 nm (blue-green), which approximates the GFP band, 825 - 870 (NIR-I), which is the emission band defined by the bandpass filter used for

the NIR imaging that follows, and the commonly used NIR-II windows 1050 - 1115 nm (NIR-II-a), 1250 - 1305 nm (NIR-II-b), 1660 - 1715 nm (NIR-II-c). Relative to the blue-green region, the NIR-I window, corresponds to a 109X increase in absorption, while the NIR-II-a/b/c windows correspond to an additional, 4/28/140X increase in absorption (440/3,101/15,339X relative to the blue-green spectra). The standard deviation of the water absorption measurements between compiled datasets was equivalent to 10.5% of the mean value over the range 700 - 1750 nm. While the increase is already substantial in the NIR-I, this range is commonly considered an optimal window where scattering and autofluorescence are strongly suppressed relative to the blue-green while the increased absorption does not cause major heating of tissue or attenuation of the excitation/emission light."

d. Presenting the camera sensitivity as "not-negligible" is an odd phrasing – perhaps amusing but not informative.

We have now changed the wording of the sentence into: A tube lens (400–1300 nm, Thorlabs TTL200MP, 200 mm focal length) was used to produce an image of the sample at 11.1x magnification on an sCMOS camera (Andor Zyla 4.2) which provides sufficient sensitivity (quantum efficiency > 10%) up to ca. 950 nm.

e. The statement about the resolution required to see nuclei is incorrect. The diameters of the nuclei are less critical than the spacing between the nuclei, which can be significantly smaller.

We thank the reviewer for this comment, which refers to line 82 of the Supplementary Information document. We have now adapted the corresponding text, which now reads:

"In any case, the difference is minor and cell nuclei (ca. 5 - 10 μm), which are the smallest structural details that we sought to resolve were easily resolvable for GFP, AF647 and CF800 in both sparsely labelled samples such as pescoids and Drosophila embryos, where the spacing between nuclei is substantial smaller (ca. 2–5 μm) as well as densely packed tissues, including the brain in the zebrafish larvae."

f. The comment on the red-shifting of the dye is telegraphic at best.

We agree with the reviewer that the comment regarding red-shifting was brief and out of place. We have moved the statement to the methods section and expanded the discussion:

The reference spectra of the NIR dyes used (AlexaFluor 800/CF800) suggest that a combination of 785 nm laser line and bandpass centered around 820 nm (for 55 nm full-width at half maximum width) would be optimal. However, both antibody/nanobody conjugation were associated with a strong redshifting of the excitation and emission spectra of the IR dyes and the 808 nm laser line and emission filter centered at 845 nm were found to be optimal. This redshifting has been observed in dyes and their conjugates and for our purposes is expected to be beneficial resulting in further decreases in scattering and autofluorescence with a small increase in absorption from water.

2. Presentation of data

The presentation of the data is not of high quality and there is minimal definition of what has been performed. Given that this is intended as a high-visibility guide on how to perform an important imaging processing, and analyzing the results this is troublesome.

a. For example, on lines 192/3, the authors state "SSIM showed a remarkable improvement in the IR2 restored images, both for zebrafish and Drosophila (mean, SD = XX, XX respectively for GFP and restored images respectively).... Even with XX, XX filled in, this sentence will not be understandable.

We thank the reviewer for noticing the missing information in the main text. We have now extended our analysis to include the same metrics of Pearson correlation, SSIM and information content for N2V. We included the values of mean and standard

deviation of all metrics in Supplementary Table 2 and restructured the paragraph to make it more readable:

Main text:

“The Pearson correlation of IR² images remained similar across all patches, while we observed a significant decrease for N2V images for both zebrafish and Drosophila samples (Figure 2 D, I and Supplementary Table 2). Notably, N2V reconstructions of both samples also displayed a lower SSIM with the ground truth (IR) (Figure 2 D, I and Supplementary Table 2). Taken together, the Pearson correlation and SSIM results suggest that, while N2V networks may learn to efficiently denoise images, they are prone to introducing artifacts (e.g., Figure 2 G, white asterisks).”

Supplementary Table 2: Image metrics relative to Figure 2.

		GFP		IR		IR ²		N2V	
		Mean	SD	Mean	SD	Mean	SD	Mean	SD
Fish data	Pearson corr	0.900	0.144	N/A	N/A	0.921	0.117	0.824	0.152
	Info content gain	N/A	N/A	1.228	0.213	1.096	0.198	0.973	0.338
	SSIM	0.865	0.071	N/A	N/A	0.929	0.046	0.766	0.097
Fly data	Pearson corr	0.820	0.153	N/A	N/A	0.750	0.212	0.531	0.442
	Info content gain	N/A	N/A	1.627	0.951	1.466	1.189	1.466	1.189
	SSIM	0.881	0.053	N/A	N/A	0.911	0.050	0.771	0.160

b. SSIM is defined as an abbreviation for structural similarity index, but not as a technique or a citation/explanation of what it does and how to use it, so readers will be

neither be able to critically evaluate the claims made by the authors nor be able to reproduce this approach for themselves.

Structural similarity index is a common metric used in image analysis to compare the similarity between two images based on their structures, or textures. To make the text easier to follow for the readers, we have now included a section in Methods describing all metrics used throughout the manuscript, and included two citations: one for the definition of the metric, and another for the code implementation.

Methods section:

“Structural Similarity Index metric

Structural similarity index is a metric used in image analysis to compare the similarity between two images⁵⁰. As opposed to easier-to-implement measures such as NRMSE and Pearson correlation, which rely on the absolute pixel values, SSIM is primarily influenced by the structures (or textures) within the images. Briefly, the SSIM between two images x and y is the multiplication of the measures of luminance, contrast and structure. Throughout this work, we have used the implementation of SSIM from the popular Python package scikit-image⁷³.”

References added:

Wang, Z., Bovik, A. C., Sheikh, H. R. & Simoncelli, E. P. Image quality assessment: from error visibility to structural similarity. IEEE Trans. Image Process. 13, 600–612 (2004).

van der Walt, S. et al. scikit-image: image processing in Python. PeerJ 2, e453 (2014).

c. Pearson’s correlation seems to have included black pixels in the analysis, which for some image times will give misleading results about the correlations in the features that are observed. I suspect that if the black pixels (or near black pixels) were to be removed as has been performed in many of the best papers, the conclusions would have been significantly different.

We thank the reviewer for this comment. Indeed, we are aware that if black pixels were to be included in the Pearson correlation computation, they would change the obtained values. Throughout the paper, we have in fact calculated Pearson correlation using

small patches of (128x128x32) pixels extracted from the 3D volumes and avoided dark regions of the image. To make this point clear to the reader, we have expanded the description of the Pearson correlation analysis throughout the text as well as added a section in the Methods.

Main text:

“To perform an objective quantification of the quality of IR staining, we extracted small volumes (patches of 128 x 128 x 32 pixels each) from the full volumes of visible and IR images avoiding dark regions of the images, and computed the pixel-wise Pearson correlation between the two (Figure 1 B, Methods).”

Methods:

“Pearson correlation coefficient

Pearson correlation between pairs of images was computed on patches of (128x128x32) pixels extracted from the whole 3D images avoiding dark regions.

For the comparison between GFP and antibody-stained IR images in Fig1B, B', we extracted patches within 20 μ m from the surface of the sample. The sample mask was computed with a manually-set threshold and the edge mask was obtained subtracting a binary erosion of the mask itself.”

d. There should be statistical evaluations of the differences that are offered, with the necessary Bonferroni multiple comparisons corrections.

For all statistical tests we obtained a statistically significant difference between datasets, that is they all gave p-values lower than the critical value, even after log-transformation of the data and Bonferroni correction. However, we have not included these values in the main text because this is not informative of the differences between the metrics values obtained for the different images for two main reasons: 1) most tests work well under low group size (<100), while our quantifications are performed on N>3000 patches and 2) commonly used statistical tests assume groups to be independent, while our data are obtained by restoration of the same patch image,

therefore an accurate estimation of co-variables is challenging. As a result, the high statistical significance obtained from our data is only indicative of the fact that the mean values of the distributions are different, something that is not surprising given the large group size. Taken together, this makes rigorous statistical comparison less powerful than direct comparison of metrics between different images. In order to provide a more in-depth analysis of these differences, we have now included a Supplementary Note and Supplementary Figure accompanying Fig 2 where a patch-by-patch difference of the metrics obtained for IR, IR2 and N2V are visualised. In the added material we quantified the increase in information content in IR2 relative to N2V, and the overall number of patches in which the information content in the IR2 images is higher than in their N2V counterpart. We then performed a statistical binomial test under the null hypothesis that N2V outperforms IR2. Overall, the probability to obtain the observed differences were well below the critical p-value (0.001), thus showing that IR2 outperforms N2V under all metrics used. This is now explained in the main text and in Supplementary Note 4, which now reads:

Main text:

“Additionally, we set out to compare the metrics on a patch-by-patch basis and perform statistical analysis under the null hypothesis that N2V outperforms our IR2 approach. This hypothesis was strongly rejected for both zebrafish and Drosophila datasets, where we observed that the few images in which N2V outperformed IR2 comprised low information content and were dominated by noise (p-value<0.001, see Supplementary Note 3 and Supplementary Figure 7 for details).”

Supplementary Note:

“Supplementary note 3. Statistical comparison IR2 vs N2V restored images.

To perform a more quantitative comparison of IR2-restored with N2V-restored images, we set out to perform a patch-by-patch statistical analysis of the metrics obtained in Fig. 2. In particular, we analyzed patches for which the information content gain of IR images relative to the GFP were higher than 1, and bootstrapped the data: we randomly extracted 100 patches to avoid biasing the statistical analysis due to the large group size (N=2484 for zebrafish and N=9982 for Drosophila). Next, we overlaid

line plots connecting metrics values for the same patch over IR, IR2 and N2V images, and color-coded them in green for patches in which IR2 values showed an improvement over IR and N2V and red for which IR2 values were outperformed by N2V (Supplementary Figure 6).

For fish data, we observed that IR2 patches had 38.0% more information content than their N2V counterpart. Overall, 71/100 of the patches had higher information content in the IR2 than in the N2V. Under the null hypothesis that the N2V approach outperforms the IR2 network, we performed a one-sided binomial test with number of trials=100 and number of successes (IR2>N2V)=69, and obtained a statistic of 0.71, corresponding to a p-value=1.6e-5. Therefore, we rejected the null hypothesis and instead showed that IR2 outperforms the N2V approach. Similar values were obtained for SSIM and Pearson Correlation (Supplementary Table 3). Similarly, we also rejected the null hypothesis for *Drosophila* data with a p-value of 2.8e-7."

Supplementary Table 3. Patch-by-Patch statistical analysis.

		N patches analyzed	% improvement IR2 over N2V	IR2 better than N2V	P-value (1-sided binomial test)
Fish data	Info content gain	100	22.2	71	1.6e-5
	SSIM	100	22.7	100	7.9e-31
	Pearson correlation	100	13.8	99	8.0e-29
Fly data	Info content gain	100	74.4	75	2.8e-7
	SSIM	100	35.5	98	4.0e-27
	Pearson correlation	100	9.2	82	3.1e-11

Supplementary Figure 7. Quantitative comparison of individual patches between IR, IR2 and N2V for zebrafish (A) and *Drosophila* samples (B). Underlying violin plots are the same as in Fig2. Overlaid lines connect metrics across approaches for the same patch, and are color-coded according to the performance of IR2 with the connecting

approach (red= IR2 worse, green=IR2 better). C), D) Distribution of absolute information content in the ground truth IR images for patches in which IR2 outperforms N2V (green) or vice versa (red). E), G) Example patches for which IR2 performs better than N2V in terms of information content gain in zebrafish and drosophila samples, respectively. F), H) Example patches for which N2V performs better than IR2 in terms of information content gain in zebrafish and drosophila samples, respectively.

3. Figures

The figures are disappointing and variable.

a. Panels have an odd mix of whisker and violin plots.

Following the reviewer's comment, we have decided to make metric visualization consistent across all main figures, and chose to use violin plots, which give important additional information regarding the distribution of the metric described for the patches, as well as summary results such as average and quartile values. Therefore, we have now replaced all whisker plots with violin plots and added the average values and the quartiles with solid and dashed lines. The relevant figures now appear as follows:

Figure 1: Highly-selective near infrared (NIR) staining and light sheet microscopy affords superior imaging at depth in tissue. A) Maximum intensity projected (MIP) z-stacks acquired for a fixed *Tg(kdrl:GFP)* (vascular marker) zebrafish larva (72 hpf) stained against GFP via conventional indirect immunostaining with AlexaFluor800 (AF800). Visible (GFP) left, NIR (AF800) right. Scale bar: 100 μ m **A')** A single superficial z-plane from a fixed *Tg(h2b:GFP)* (nuclear marker) zebrafish larva/embryo (96 hpf) stained against GFP via nanobody-conjugate CF800. Visible (GFP) left, NIR (CF800) right. Scale bar: 100 μ m **B, B')** Selected superficial patches shown by the dashed boxes in A, A' respectively (visible (GFP) top, NIR

(AF800/CF800) bottom) and pixelwise correlation plots for all 125 extracted patches. Scale bar: 5 μ m **C)** Multiple deeper z-planes acquired for the same nuclear marker (h2b) zebrafish embryo/larva shown in B. Scale bar: 100 μ m **D)** Pearson correlation and structural similarity index measure (SSIM) for the full z-stacks acquired for the vascular (kdr1) and nuclear (h2b) marker zebrafish from A, A' respectively. Dash lines represent 25/50/75th quartiles. **E)** Selected deeper patches for the vascular (kdr1) and nuclear (h2b) marker zebrafish from A, A' respectively. Scale bar: 5 μ m **F)** Pearson correlation and SSIM for all patches extracted at different z-planes from the full z-stacks of the vascular (kdr1) and nuclear (h2b) marker zebrafish. The z-depth provided is the maximum z-depth into tissue for each image in the stack. The uncertainty envelope is given by standard deviation. **G)** The information content gain (see Methods), between the visible (GFP) and IR channels (I_{IR}/I_{GFP}) for all patches extracted at different z-planes from the full z-stacks of the vascular (kdr1) and nuclear (h2b) marker zebrafish. The uncertainty envelope is given by standard deviation.

Figure 2: Infrared-mediated Image Restoration (IR²) improves image quality of degraded GFP images. **A)** Single GFP and IR images (left columns) extracted at increasing detection depth in a 96 hpf Tg(h2b:GFP) zebrafish larva restored with either IR² or Noise2Void (N2V), (right columns). Scale bar: 100µm **B)** Example patches for the same zebrafish dataset shown in A) arranged for increasing cell density. Scale bar: 5µm **C)** Violin plots of information content gain (relative to GFP) in patches extracted from the ground truth (IR, dark red), IR²-reconstructed (IRIR, orange) and N2V-reconstructed (N2V, yellow) images. Vertical gray lines indicate standard deviation. **D)** Pearson correlation and SSIM obtained for patches extracted in the GFP, IR²- and N2V-restored images when compared with the ground truth image (IR). **E)** SSIM relative to IR image as a function of detection depth for patches extracted throughout the sample. **F)** Single Z planes at increasing detection depth for a Tg(His2AV-GFP) fly larva (8 hpf) extracted from the input (GFP) and ground truth image (IR), as well as from restored images obtained from IR² and Noise2Void. Scale bar: 100µm **G)** Example patches for the same fly dataset shown in F). White asterisks indicate patches where artifacts were introduced or features were not reconstructed by the Noise2Void network. Scale bar: 5µm **H)** Violinplot of information content gain (relative to GFP) in patches extracted from the ground truth (IR, dark red), IR²-reconstructed (IRIR, orange) and N2V-reconstructed (N2V, yellow) images. Vertical gray lines indicate standard deviation. **I)** Pearson correlation and SSIM relative to IR image, for GFP, infrared-mediated and Noise2Void reconstructions. **L)** SSIM relative to NIR images as a function of detection depth. In all violin plots, dashed lines represent 25/50/75th quartiles.

Figure 3: The quality of IR² images is robust over large developmental intervals.

A) Representative GFP and IR images of a single Z-plane in a full 3D stack of *Tg(kdrl:GFP)* zebrafish larvae at 2, 3 and 4 days post fertilization. Scale bar: 100 μ m

B) Image reconstructions obtained from IR² models trained with images from 2, 3 and 4 dpf zebrafish larvae.

C) Quantification of image quality as measured with the normalized root mean squared error (NRMSE) and SSIM from the GFP image as well as the images obtained after IR² reconstruction. Values were averaged over all patches extracted from 4 different samples per age group. Non-transparent violin plots represent the values from the GFP images and the images reconstructed from the IR² network matching the zebrafish age. Transparent violin plots represent reconstruction performed with networks from other developmental stages. Horizontal solid and dashed lines represent the mean and quartile values, respectively.

b. Figures present Pearson's analyses with the scaling dominated by the black (0,0) pixels so that patterns, if they are there, cannot be discerned.

This comment is addressed in Point 2c of the same reviewer.

c. The comparison of images with depth assumes that a reader can judge differences in the images. A difference map should be offered, with a color scale and an offset that makes it simple for the reader to judge errors of commission and omission.

We thank the reviewer for this useful suggestion. We have now included Supplementary Figure 5 with "difference maps" for the images shown in Figure 2, as well as a section in Methods to explain the procedure to obtain such maps, which reads:

"Normalized difference map

The normalized difference maps shown in Supplementary Figure 6 are obtained by normalizing the GFP, IR, IR2 and N2V with their respective 0.3 and 99.9 percentiles, to obtain images with identical dynamic range. Next, the absolute values of the difference between pairs of images was computed and shown. For visualization, we randomly chose patches in which the IR image showed significant image quality improvement compared to the GFP image (that is, patches in which the IR image had an information content gain higher than 1.2)."

Supplementary Figure 6. **A)** IR, GFP, IR2- and N2V-restored images extracted at increasing detection depth in a 96 hpf *Tg(h2b:GFP)* zebrafish larva, shown side by side with their respective difference map relative to the ground truth IR image. Scale bar: 100 μm **B)** Example patches and relative difference maps for the same zebrafish sample shown in A) arranged for increasing average normalized root mean squared error in the IR2 patch. Scale bar: 5 μm . **C)** IR, GFP, IR2- and N2V-restored images extracted at increasing detection depth in a 8 hpf *Tg(His2AV-GFP)* drosophila larva, shown side by side with their respective difference map relative to the ground truth IR image. Scale bar: 100 μm . **D)** Example patches and relative difference maps for the same drosophila sample shown in A) arranged for increasing average normalized root mean squared error in the IR2 patch. Scale bar: 5 μm .

d. The N2V denoising neural network outperforms the authors' technique in some ways (by SSIM plot), and without the difference map requested in 3c, the comments from the authors on its poor performance are not supported (the only metrics that are offered show it is superior).

We thank the reviewer for this comment. While N2V is indeed comparable in terms of info content gain (not SSIM) only for Drosophila dataset, we have highlighted in the main text and by means of SSIM metrics how this behaviour is due to the introduction of artefacts. The main text now reads as follows:

"To benchmark IR² against current restoration methods, we restored both zebrafish and drosophila images using Noise2Void, a self-supervised deep learning algorithm for image denoising²¹, and visually compared the absolute difference map for individual planes and example patches (See Methods and Supplementary Figures 6). To quantify performance, we computed the gain in information content of IR, IR²- and N2V-reconstructed images relative to the input GFP image (see Methods for a definition of information content). We observed that the information content gain of N2V did not outperform that of IR², and actually reduced information content (information content < 1) for zebrafish samples, suggesting that the degradation of the input GFP images was not dominated by noise (Figure 2 C, H). Conversely, we observed that the IR² images had significantly higher information content gain, suggesting that scattering at depth was the major source of degradation, and could efficiently be restored using IR² networks (Figure 2 C, H and Supplementary Table 2). The Pearson correlation of IR² images remained similar across all patches, while we observed a significant decrease for N2V images for both zebrafish and Drosophila samples (Figure 2 D, I and Supplementary Table 2). Notably, N2V reconstructions of both samples also displayed a lower SSIM with the ground truth (IR) (Figure 2 D, I and Supplementary Table 2). Taken together, the Pearson correlation and SSIM results suggest that, while N2V networks may learn to efficiently denoise images, they are prone to introducing artifacts (e.g., Figure 2 G, white asterisks)."

To make this point clearer to the reader, we have expanded our analysis and performed a more direct comparison of reconstructed patches as highlighted in point 2D from the same reviewer.

e. The figure of IR-dye (AF700) staining is very bad. If the figure is an accurate rendering, it is a failed experiment and the comparison offered in Supp Fig 1 is meaningless.

We agree with the reviewer's assessment that this is a failed experiment. Nevertheless, we believe it warrants inclusion for two reasons. First, it provides a useful demonstration that certain antibody/dye combinations result in poor selectivity that cannot be predicted *a priori*. Second, the poor quality of the staining, together with the quantification shown in the same figure, demonstrates that the Pearson correlation coefficient is a good metric to quantify staining selectivity.

f. There are claims made about the axial and lateral chromatic aberrations, which should be shown not just dismissed.

We welcome the opportunity to delve deeper into the chromatic performance of the system. In addition to the existing text in the supplementary note 1 which we believe goes some way to covering the reviewers point about axial chromatic aberrations (through a description of the focus correction and PSF dimensions), we have added the following additional discussion and a figure, which shows the registration of patches during the IR2 processing pipeline and fully aligns the volumes in 3D for subsequent training.

Modifications to Supplementary Note 1 (Previously Supplementary Note 2):

"Despite the refocus step some axial misalignment of the individual colors is still possible and may vary depending on depth in tissue or the depth inside the agarose bead column for which the refocus calibration is made. Moreover, the refocusing does not correct for lateral chromatic aberrations. Since the IR2 training is reliant on the 1:1 correspondence between different color channels, it is necessary that residual misalignments are corrected for. Any residual axial/lateral chromatic aberrations are accounted for via a registration step (see "Deep Learning" in Methods). As a demonstration, a two-color fluorescent bead sample is shown in Supplementary Figure 1. As shown in A - C there is noticeable misalignment between the two color channels.

The misalignment is smaller in z highlighting that the refocus scheme broadly corrects for axial color. Following registration the beads are well aligned (D - E)."

Supplementary Figure 1: Multi-colour patch registration. A maximum intensity projection from a stack of fluorescent beads: cyan: 488 nm excitation, 525/50 nm emission (band centre/full-width at half maximum bandwidth), magenta: 640 nm excitation, 697/60 nm emission. A: The centre $200 \times 200 \mu\text{m}$ of the full imaging field of view (scale bar = $50 \mu\text{m}$). B - E one patch extracted from the full volume: $18.7 \times 18.7 \times 39.7 \mu\text{m}$ shown as maximum intensity projections over the excluded axis (scale bars = $5 \mu\text{m}$). B/D xy views C/E xz views. B/C pre-registration. D/E post registration.

In summary, I am very supportive of this work and urge the authors to make it clear, accessible, bolstered by data that is offered easily, not on request.

Reviewer #2:

Remarks to the Author:

The authors proposed the use of a deep learning method (convolutional neural networks) to restore deep-tissue contrast in GFP-based visible imaging using paired final-state datasets acquired using NIR-I dye. Frankly, a similar idea has been published in 2021 (<https://doi.org/10.1073/pnas.2021446118>), in which they used a deep-learning-based approach to transform a blurred NIR-I (800-1000 nm) image to a much higher-clarity image like in NIR-IIb (1500-1700 nm) to remove the scattering effect in the NIR-I window. On the other hand, the authors used a common convolutional neural network from the published CARE package (Ref.19, Nat. Methods 15, 1090–1097 (2018)). The idea and the methods in this manuscript are not new. So I do not recommend publication except the authors give more evidence to show its novelty.

The following are some questions/comments about this manuscript.

1. The idea of this manuscript is similar to a published article (<https://doi.org/10.1073/pnas.2021446118>) which enhanced NIR-I (800-1000 nm) imaging by NIR-IIb (1500-1700 nm) imaging using deep learning methods. What is the advantage of the authors' methods?

We thank the reviewer for highlighting this article, which had escaped our attention. We believe that our method has several benefits and have added the following discussion:

"Unlike others who have used convolutional neural networks to restore image quality on the basis of reduced scattering at longer wavelengths⁵⁹, IR² offers the following advantages. First, imaging is performed using well-corrected immersion optics with a 3D PSF that is as much as 250X smaller by volume based on the measured resolution in either case³⁸. Secondly, IR² relies on genetically-encoded fluorophores and their

cognate antibody/nanobody-tagged NIR dyes thus ensuring molecularly-precise correspondence between the ground truth and degraded data. Furthermore, the labeling scheme achieves cell-permeability and so one is not limited to intravenous injection as a delivery mechanism. Coupled with the selectivity noted, one may image arbitrary tissues and sub-cellular components rather than being limited to imaging the vasculature through non-selective dispersion of the dye in the bloodstream or to cell-surface receptors where circulating dye may bind. Thirdly, IR² has been developed specifically for the restoration of time-lapse images and has been shown to be robust across wide developmental windows. From a practical implementation point-of-view, IR² relies on deep learning networks that are easy to utilize using a widely used, well established deep learning library (CARE)²⁰. Implementations for this and similar deep learning libraries exist for a variety of image analysis ecosystems, including FIJI^{60,61}, Python and Napari plugins^{62,63}. Fourthly, IR² uses only widespread commercially available dyes and inexpensive silicon-based cameras, rather than custom synthesized dyes and costly InGaAs models. Taken together, these aspects open the NIR/deep learning toolbox to cell and developmental biologists wishing to push live-imaging deeper into tissue. Conversely, IR² is more limited in terms of maximum imaging depth owing to the NIR-I vs. NIR-II operating range. Nevertheless, for imaging with cellular resolution in mm-sized models the loss of spatial resolution at longer wavelengths likely dominates in a tradeoff since the optical penetration in the NIR-I is typically sufficient for in toto imaging of small embryos/larva."

2. The optical diffraction limit confined resolution of IR imaging is lower than visible/GFP imaging. Did the authors consider this influence when they performed IR2 (infrared-mediated image restoration)?

The reviewer is quite correct with this assertion. We did consider this influence, previously this was described only in the supplementary information but we have noted in the main text that the assertion, while true, often has no practical influence since diffraction limited resolution in the visible spectrum is only achievable in superficial regions and under the sampling conditions typical to light sheet imaging the spatial resolution is rather limited by the Nyquist-Shannon criterion. Even so, when observing finer/denser features such as cell nuclei we have ensured that the sampling was

appropriate to observe the effect of loss of spatial resolution. We have added the following discussion and highlight the relevant section of the supplementary information

Main text:

"It is worth noting that the achievable diffraction-limited spatial resolution scales inversely with wavelength. However, the diffraction-limited spatial resolution is really only achieved within a cell layer or two of the surface. As such, it is a common practice in light sheet microscopy to undersample with respect to the Nyquist-Shannon sampling criterion to maximize field of view. The data of Figure 1 A - B were collected under undersampled conditions for both the GFP and NIR spectral regions. For the denser nuclear data and all data that follows the spatial sampling was increased to potentially allow finer features to be resolved. In fact, the decrease in spatial resolution in the NIR relative to GFP was less than expected (31/17% in xy/z) from a direct comparison of imaging wavelength as described in Supplementary Note 1. The degraded resolution owing to the light-tissue interaction will dominate in any case for the deeply located tissues of interest."

Supplement:

*"To explore whether this was apparent, bead stacks were analysed using PSFj⁴. The lower bound for the lateral PSF FWHM for excitation/emission centers of 488/525 nm, 640/697 nm, 808/845 nm was found to be 741 ± 14 , 797 ± 15 and 974 ± 18 nm, which are equivalent to maximum NAs of 0.44, 0.54, 0.54 respectively (from the Rayleigh criterion), demonstrating that the optical performance is best in the far-red - NIR as expected and approaches the theoretical value of 0.6. Although the lateral PSF for the GFP equivalent is slightly narrower, we note that this will be apparent only in extremely superficial regions. In any case, the difference is minor and cell nuclei (ca. 5 - 10 μm), which are the smallest structural details that we sought to resolve were easily resolvable for GFP, AF647 and CF800 in both sparsely labelled samples such as *pescoids* and *Drosophila* embryos, where the spacing between nuclei is substantial smaller (ca. 2-5 μm) as well as densely packed tissues, including the brain in the*

zebrafish larvae. The axial PSF FWHM is also similar for the three imaging bands: 5.46 ± 0.22 , 5.36 ± 0.25 and $6.38 \pm 0.41 \mu\text{m}$ for 488/525 nm, 640/697 nm, 808/845 nm.”

3. The authors used a common convolutional neural network from the published CARE package (Ref.19). What is new?

The reviewer is correct in stating that in our work we used a common convolutional neural network from a previously published package. We strongly believe that this is a rather important and positive point of our work, as U-Net convolutional neural networks, thanks to their easy-to-train architecture, are at date one of the most widely used deep learning implementations in the field of bioimage analysis. At the same time, CARE, a modern yet well-established implementation of such networks, is available for the two different ecosystems (Python and FIJI/ImageJ) that are used in nearly any research lab performing quantitative analysis of biological images. We agree with the reviewer that this is an important point for the reader, and we have added the following sentences to the discussion:

“From a practical implementation point-of-view, IR² relies on deep learning networks that are easy to utilize using a widely used, well established deep learning library (CARE)²⁰. Implementations for this and similar deep learning libraries exist for a variety of image analysis ecosystems, including FIJI^{60 61}, Python and Napari plugins^{62,63}. Fourthly, IR² uses only widespread commercially available dyes and inexpensive silicon-based cameras, rather than custom synthesized dyes and costly InGaAs models. Taken together, these aspects open the NIR/deep learning toolbox to cell and developmental biologists wishing to push live-imaging deeper into tissue.”

4. According to the lasers and fluorescence bandpass filters in the Methods, the applied wavelength window is ~ 400-800 nm. However, the authors used “infrared” in the title and all over the manuscript. It is not accurate. “Near-infrared” or “NIR-I” would be better.

The reviewer is quite correct in their assessment that NIR offers a more accurate reflection of the wavelength window used (the emission bandpasses define a working window of 500 - 875 nm). We believe that the title offers a general description of the basis of contrast enhancement and as such is suitable. However, we agree that the general reader may not appreciate the differences between the NIR (NIR-I/II) and the mid/long-wave infrared associated with thermal imaging. We have changed the text throughout to refer to NIR and near infrared specifically. In some places when referring to specific spectral windows we also refer to NIR-I and NIR-II, which are explained in the text. For example:

“Conversely, near infrared (NIR, 750 - 1750 nm, comprising NIR-I 750 - 1000 nm, NIR-II 1000 - 1750 nm) light maintains its directional propagation deeper into tissue²⁷, as leveraged by two/three-photon microscopy, which relies on the absorption of multiple NIR photons to excite fluorophores with emission spectra in the visible range.”

5. Line 66, the authors claimed that “but suitable NIR fluorophores are incompatible with live imaging (a discussion of infrared imaging is given in Supplementary Note 1).” This claim is quite confusing as a lot of NIR dyes have been used for live imaging. Some of them such as ICG has been approved by FDA for human imaging.

In Supplementary Note 1, for multiphoton microscopy, “these delicate samples are damaged by the intensely pulsed light and develop too quickly for their dynamic processes to be captured via serially point-scanned schemes.” Recently many groups have demonstrated the laser power they used is safe, such as the work done by Prof. Chris Xu (Nature Methods 15, 789–792 (2018)). Other groups also developed ultrafast two-photon fluorescence microscopy with kilohertz speed (Nature Methods 17, 287–290).

Many thanks to the reviewer for this point. We have added substantial additional discussion around NIR dyes and imaging as well as multiphoton microscopy and are very keen to add additional discussion of this important topic. Discussion of highly advanced two-photon implementations such as the article noted as well as (Zhang, T. *et al.* Kilohertz two-photon brain imaging in awake mice. *Nat. Methods* 16, 1119–1122 (2019)) is outside the scope of this article. Although they provide an exciting route to

rapid multiphoton imaging, the underlying relationships between temporal resolution, evolved signal and power deposited are unchanged and we do not believe these techniques fundamentally change the suitability of multiphoton techniques in delicate developing embryos and larvae. We have added the following passages (please see manuscript for full referencing of the text):

“Conversely, near infrared (NIR, 750 - 1750 nm, comprising NIR-I 750 - 1000 nm, NIR-II 1000 - 1750 nm) light maintains its directional propagation deeper into tissue²⁷, as leveraged by two/three-photon microscopy, which relies on the absorption of multiple NIR photons to excite fluorophores with emission spectra in the visible range. Multiphoton microscopy²⁸ provides sufficient depth penetration for in toto imaging of small animal models such as embryonic/larval zebrafish²⁹ and drosophila³⁰. However, while the energy deposited and temperature changes induced by the intensely pulsed light have been shown to be safe for imaging small sub-volumes of the brains of adult zebrafish³¹ and mouse^{32,33}, phototoxic effects take hold long before physical damage is noticeable^{34,35} and for in toto imaging of delicate developing embryos and larva, the deleterious influence of multiphoton imaging is often apparent despite efforts to reduce photodamage³⁰. Furthermore, serially point-scanned schemes are typically too slow to capture developmental processes. Nevertheless, multiphoton techniques remain a powerful tool in the light microscopy arsenal for intravital imaging and remain the gold-standard for deep tissue fluorescence imaging.

For deep tissue imaging in developing embryos it is desirable to employ techniques which benefit from the penetration at NIR wavelengths, coupled with the speed and low intensity requirements of camera-based widefield techniques. However, single-photon NIR schemes are limited by a comparative paucity of live-imaging compatible fluorophores. Although dyes such as indocyanine green are FDA-approved for use in humans, they^{36,37} and other large-molecule^{38,39}, macromolecular⁴⁰ or nanoparticle dyes⁴¹ are not cell-permeable. The imaging of developmental dynamics in small embryos however, requires that subcellular components or populations of cells can be induced to express fluorescent proteins or be selectively labeled. NIR FPs only extend partially into the NIR-I with their emission spectra, require visible excitation, and suffer from being dim, weakly photostable, often dimeric and require biliverdin as a

chromophoric co-factor⁴². The self-labeling Halo- and SNAP- tagging systems provide the required selectivity and have been used with red dyes to image developing embryos⁴³ but are limited for NIR imaging by the cell-impermeability of NIR dyes. Likewise, these highly-specialized genetically-encoded tools are not widely available in animal models, limiting their applicability."

6. The authors claimed, "Although a light sheet microscope compatible with dyes emitting up to 1700 nm has been reported²⁷, absorption from water increases > 100x compared to an optimum window at 800 - 950 nm²⁶ contributing to heating and attenuation of the excitation/emission light". How the authors did this calculation? Have the authors measured the water absorption by themselves? "100x" is much larger than the reviewer's measurements.

We thank the reviewer for raising this topic, which was only touched upon in the original submission and for the opportunity to discuss further. We have compiled water absorption data from three widely used studies, which are in good agreement. We have then defined several spectral bands commonly used for biological imaging in the visible/NIR-I/NIR-II. We have thus shown that the increased absorption from the NIR-I to the NIR-II is of the order 4 - 140x for the various imaging windows and much higher for the NIR-II peak at around 1450 nm (ca. 750x for an equally broad spectral window of 55 nm). Previously we had taken an average over the entire spectral region but believe this approach with clearly defined windows is more informative and useful for others assessing NIR-I vs. NIR-II imaging. Please see the following discussion in the Results section and the manuscript for literature references:

"Although a light sheet microscope compatible with dyes emitting up to 1700 nm has been reported³⁶, absorption from water increases substantially at longer NIR wavelengths. We compiled water absorption data⁴⁴⁻⁴⁶ and define 5 spectral bands each of 55 nm width: 500 - 555 nm (blue-green), which approximates the GFP band, 825 - 870 (NIR-I), which is the emission band defined by the bandpass filter used for the NIR imaging that follows, and the commonly used NIR-II windows 1050 - 1115 nm (NIR-II-a), 1250 - 1305 nm (NIR-II-b), 1660 - 1715 nm (NIR-II-c). Relative to the blue-green region, the NIR-I window, corresponds to a 109x increase in absorption, while

the NIR-II-a/b/c windows correspond to an additional, 4/28/140× increase in absorption (440/3,101/15,339× relative to the blue-green spectra). The standard deviation of the water absorption measurements between compiled datasets was equivalent to 10.5% of the mean value over the range 700 - 1750 nm. While the increase is already substantial in the NIR-I, this range is commonly considered an optimal window where scattering and autofluorescence are strongly suppressed relative to the blue-green while the increased absorption does not cause major heating of tissue or attenuation of the excitation/emission light."

7. Line 98, "overfixation of the tissues may decrease the relative brightness of GFP and background autofluorescence." Do the authors mean "increase" background autofluorescence?

We thank the reviewer for noticing this typo in the text. Indeed, the effect of paraformaldehyde in overfixed samples is to decrease GFP brightness as well as increase autofluorescence. We have modified the sentence accordingly, which now reads:

"overfixation of the tissues may decrease the relative brightness of GFP and increase background autofluorescence⁴⁷."

8. Line 101, "A discussion of the optimization of the protocols used in this work is provided in Supplementary Note 1." I did not see the mentioned discussion in Supplementary Note 1.

Indeed, the discussion on the strategies used for fixation, permeabilization and staining are discussed in Supplementary Note 2. Thanks to the reviewer for pointing this out.

9. Why the GFP results and IR results in Fig. 1A and 1A' do not show an apparent difference but the Fig. 1C shows a difference?

As the reviewer pointed out, images shown in Fig1 A, A' do not provide an appreciable difference between the GFP and IR images. The first is the image of a vasculature-labelled sample, which is a notably easy-to-image fish sample, for which the IR improvement is not appreciable, since the information is mainly at low spatial frequencies, which remain in the degraded images. The second is a single Z-plane of a more challenging nuclear-labeled fish sample. We included these images with the aim of showing that the experimental protocol used provided a selective staining against the fluorescence protein of interest (GFP). These issues are described in detail in Supplementary Note 2, Supplementary Table 1, Supplementary Figure 2 and 3.

Regarding Fig1C, this is a direct visual comparison of GFP vs IR images at different depths, and the first example in which IR images provide an improved image quality over GFP images.

10. What is the exposure difference between the GFP images and IR images in Fig. 1E'? It looks like the GFP used a shorter exposure time. What are the exposure time and frame rate of the images in this manuscript? What is the fluence of the laser? Are they safe for small animals? Details are needed.

We thank the reviewer for raising this important point. Indeed, in order to avoid our analysis being dependent on the overall brightness of the images, we chose imaging conditions for GFP and IR to make sure that the brightness of the images were similar. To this aim, for the fixed and stained samples, given the lower brightness of IR dyes such as CF800, we chose an exposure time of 100 ms for both GFP and IR, and a laser power of 5-20mW for GFP and 10-50mW for CF800. In the case of live animals, which were imaged only in the GFP channel, we reduced the exposure time to 20 ms to avoid phototoxicity and achieve an appropriate temporal resolution.

A detailed description of the imaging conditions use is now given in the Methods section, and reads as follows:

"Sample imaging

Imaging of zebrafish and Drosophila fixed samples was performed on IR-mSPIM (see Methods for details) using 488 nm (GFP) and 808 nm (CF800 dye) excitation

wavelengths. The laser powers were chosen to make sure that CF800 images would have an absolute brightness comparable to the GFP ones. For the zebrafish samples, stacks were generated acquiring a Z plane every 5 μm using laser powers of 5-20 mW (GFP) and 10-50 mW (CF800) and exposure times of 100 ms for both. For the *Drosophila* samples, volume stacks were generated acquiring a Z plane every 2.5 μm using laser powers of 5-20 mW (GFP) and 10-50 mW (CF800) and exposure times of 100 ms for both.

Live zebrafish and *Drosophila* samples were imaged using similar GFP settings, except for the exposure time, which was set at 20 ms to avoid phototoxicity.

Imaging of fixed pescoids was performed on a commercial light sheet system (Luxendo MuVi-SPIM, Olympus 20X/1.0NA detection objective, 16.7X effective magnification, 0.39 $\mu\text{m}/\text{pixel}$), using 10 mW laser power and 50 ms exposure time for both GFP (488 nm wavelength) and AlexaFluor647 (642 nm wavelength). Imaging of live pescoids was performed on the same microscope using 3.5 mW laser power and 50 ms exposure time. In both cases, stacks were generated acquiring a Z plane every 2 μm . Live images acquired by the two opposing camera views were registered and fused using the Image Processor module of the Luxendo software (Luxendo processor software v3.0).

For all images, 3D reconstructions and movies were performed using the FIJI plugin 3DScript⁷².

11. Line 170, "Next, the restoration network was trained (see Supplementary Note/Methods) using the GFP and IR images of the nuclear marker as degraded and ground truth datasets, respectively." I did not see the restoration network or training methods in the Supplementary Note.

We thank the reviewer for highlighting the incorrect information in the main text. The description of the deep learning approach is given in the Methods section and in Supplementary Figure 4. To make this clearer to the reader, we rephrased the main text, which now reads:

"Next, we used a U-Net deep learning network⁵⁵ and trained it using the GFP and IR images of the nuclear marker as degraded and ground truth datasets, respectively (see Methods)."

12. "The information content gain of an image 1 relative to an image 2, is defined as the ratio between the information content values of the two images." What are the "information content values"?

Information content is measured as the Shannon entropy of the direct cosine-transformed image. A description is provided in Methods, and we have rephrased the section, which now reads:

"Entropy-based information content

Similar to previous approaches^{51,74}, we measured image information content (IIC) by computing the Shannon entropy of the discrete cosine transform of the image patch:

$$IIC = - \sum_{i,j} p_{i,j} \cdot \ln(p_{i,j})$$

$$p_{i,j} = \frac{F(i,j)}{\sum_{i,j} F(i,j)}$$

$$F(i,j) = \frac{DCT(I)^2}{N^2}$$

Where N represents the size of the patch and DCT is the discrete cosine transform of the image patch I.

Throughout the text, the information content gain of an image 1 relative to an image 2, is defined as the ratio between the image information content values of the two images."

13. In Fig. 2G, I did not observe the obvious difference between GFP, IR and IR2 images.

We thank the reviewer for raising this topic, which was also pointed out by Reviewer 1. We have now included Supplementary Figure 5 showing difference maps for restored images, including those shown in Fig 2A,B,F,G, as well as a section in the Methods explaining the procedure followed to obtain such maps:

“Normalized difference map

The normalized difference maps shown in Supplementary Figure 6 are obtained by normalizing the GFP, IR, IR2 and N2V with their respective 0.3 and 99.9 percentiles, to obtain images with identical dynamic range. Next, the absolute values of the difference between pairs of images was computed and shown. For visualization, we randomly chose patches in which the IR image showed significant image quality improvement compared to the GFP image (that is, patches in which the IR image had an information content gain higher than 1.2).”

Supplementary Figure 6. A) IR, GFP, IR2- and N2V-restored images extracted at increasing detection depth in a 96 hpf Tg(h2b:GFP) zebrafish larva, shown side by side with their respective difference map relative to the ground truth IR image. Scale bar: 100 μm **B)** Example patches and relative difference maps for the same zebrafish sample shown in A) arranged for increasing average normalized root mean squared error in the IR2 patch. Scale bar: 5 μm . **C)** IR, GFP, IR2- and N2V-restored images extracted at increasing detection depth in a 8 hpf Tg(His2AV-GFP) drosophila larva, shown side by side with their respective difference map relative to the ground truth IR image. Scale bar: 100 μm **D)** Example patches and relative difference maps for the same drosophila sample shown in A) arranged for increasing average normalized root mean squared error in the IR2 patch. Scale bar: 5 μm .

We also would like to take the opportunity to discuss this topic further, also as a reply to the reviewer's general remark about the need to give more evidence of the novelty of our work. By their nature, deep learning models are capable of detecting underlying differences that are not always visible by the human eye, but that could instead become evident when a quantification pipeline is performed over the input and restored images. For this reason, we have now included in the manuscript a new set of experiments, performed using embryonic organoids, that aim at highlighting how IR2-restored images can greatly aid a common developmental biology task: namely, cell lineage reconstruction. In this specific application, we found that cell lineages obtained from IR2-reconstructed images of developing zebrafish embryonic organoids were superior to the ones obtained from input GFP images in several ways. First, a greater number of cells were detected overall throughout the dataset. Second, the additional cells were distributed mostly in the inner part of the sample, where the quality of GFP images degraded. Thirdly, because of the superior cell segmentation, we could obtain longer, high fidelity lineages. This is now explained in the main text and in the main Figure 5:

“Following demonstration of the image quality improvement achieved by IR² on time-lapse live imaging, we next sought to demonstrate how this approach can aid a quantitative analysis common in biological imaging: namely, cell lineage reconstruction. To this end, we used a novel organoid model system, termed pescoids,

which consist of zebrafish embryonic explant⁵⁶. Furthermore, to explore the applicability of our approach using more widely available optical and biochemical tools, we used a commercial antibody conjugated with a dye in the far-red (antiGFP:AlexaFluor647) and a commercially-available multi-view light sheet microscope (MuVi SPIM, see Methods). The images showed a clear increase in image contrast at depth (Supplementary Figure 10). We then trained an IR² network and applied it to timelapse data of pescoids⁵⁷(see Methods for details on mounting and imaging conditions). We observed an increased contrast in the IR² images compared to live GFP data (Figure 5 A and Supplementary Movie 2), also quantitatively confirmed by plot profile analysis (Figure 5 B), that persists for all time points (Supplementary Movie 3). Next, we used well-established software based on a Gaussian Mixture Model to perform cell tracking on both GFP and IR² timelapse datasets (TGMM)⁵⁸. We observed that TGMM was able to detect a substantially larger number of cells in the IR² images compared to the GFP data at all time points, suggesting that the increased contrast obtained in the reconstructed data greatly aids cell detection (Figure 5 C). Furthermore, as IR² images are expected to provide significant image quality gain deeper in the sample tissues, we also observed that TGMM was capable of detecting cells in the IR² images in regions where the GFP dataset showed only few detected cells (Figure 5 D). As a consequence of the improved cell detection at all time points, we observed a clear benefit in whole lineage reconstruction in IR² images, where we obtained substantially longer tracks (Figure 5 E). Evidently, the higher fidelity on track reconstruction from IR² images is a direct benefit of increased image quality, as observed when visualizing a random subset of all the cell tracks obtained from the reconstructed data in both datasets (Figure 5 F, G)."

Figure 5: Image restoration of developing pescoid. *A)* Maximum intensity projections (MIP) of Z planes between 240 and 280 μm deep inside a pescoid (cartoon) developing over the course of approximately 10 hours. Top and bottom panels represent the live GFP images and the images reconstructed using IR^2 , respectively. Scale bar: 50 μm . *B)* Top: A single Z plane from the volume acquired at the first time point, where GFP and IR^2 images have been alternated in vertical stripes. Bottom: the plot profile along the white dashed line in the GFP (blue) and IR^2 images (orange). Scale bar: 50 μm . *C)* Number of cells detected over time in the GFP (blue) and IR^2 (orange) volumes. *D)* Number of cells detected in GFP (blue) and IR^2 (orange) images at the first time point as a function of the distance from the center of mass (inset). *E)* Distribution of track lengths in GFP (blue) and IR^2 (orange) timelapse

images. F) Example of images from cell tracks in GFP and IR2 timelapse. Scale bar: 10 μm . G) 3D representation of tracks longer than 50 timepoints, color coded for time (light blue is early timepoints, violet is late timepoints). Black lines: tracks represented in F). Scale bar: 50 μm .

14. Line 193, "(mean, SD = XX, XX respectively for GFP and restored images respectively)," What is "XX"?

We thank the reviewer for noticing the missing information, which was also pointed out by reviewer 1. We have now extended the analysis, rephrased the whole paragraph and added the values in Supplementary Table 2. In particular, we have now included the same metrics of Pearson correlation, SSIM and information content for N2V. We included the values of mean and standard deviation of all metrics in Supplementary Table 2 and restructured the paragraph to make it more readable:

Main text:

"The Pearson correlation of IR² images remained similar across all patches, while we observed a significant decrease for N2V images for both zebrafish and Drosophila samples (Figure 2 D, I and Supplementary Table 2). Notably, N2V reconstructions of both samples also displayed a lower SSIM with the ground truth (IR) (Figure 2 D, I and Supplementary Table 2). Taken together, the Pearson correlation and SSIM results suggest that, while N2V networks may learn to efficiently denoise images, they are prone to introducing artifacts (e.g., Figure 2 G, white asterisks)."

Supplementary Table 2. Image metrics relative to Figure 2.

		GFP		IR		IR2		N2V	
		Mean	SD	Mean	SD	Mean	SD	Mean	SD
Fish data	Pears on corr	0.900	0.144	N/A	N/A	0.921	0.117	0.824	0.152
	Info conte nt gain	N/A	N/A	1.228	0.213	1.096	0.198	0.973	0.338

	SSIM	0.865	0.071	N/A	N/A	0.929	0.046	0.766	0.097
Fly data	Pears on corr	0.820	0.153	N/A	N/A	0.750	0.212	0.531	0.442
	Info content gain	N/A	N/A	1.627	0.951	1.466	1.189	1.466	1.189
	SSIM	0.881	0.053	N/A	N/A	0.911	0.050	0.771	0.160

15. For Fig. 3A, do the authors have IR raw images at 2-, 3- and 4-days post fertilization for comparison or used as ground truth?

We thank the reviewer for the comment, and take the opportunity to highlight that the IR raw images for the zebrafish samples restored in Figure 3 are shown in Panel A, second column. Regarding the ground truth dataset, we fixed and stained 5 fish per developmental stage, used the datasets from 1 zebrafish to train the deep learning networks (training and validation set), and restored the other 4 zebrafish samples (test set). The 4 days old fixed and stained animal used for network training are shown in Figure 1 Panel C.

Decision Letter, first revision:

23rd Aug 2023

Dear Jan,

Thank you for submitting your revised manuscript "Image restoration of degraded time-lapse microscopy data mediated by infrared-imaging." (N METH-A50903A-Z). It has now been seen by the original referees and their comments are below. The reviewers find that the paper has improved in revision, and therefore we'll be happy in principle to publish it in Nature Methods, pending minor revisions to satisfy the referees' final requests and to comply with our editorial and formatting guidelines.

We thought your revision plan was very fair and we ask that you update along those lines. We do ask that you include relevant citations backing your illumination intensities used for the live animal imaging and make necessary updates to improve the usability of shared code.

TRANSPARENT PEER REVIEW

Nature Methods offers a transparent peer review option for new original research manuscripts submitted from 17th February 2021. We encourage increased transparency in peer review by publishing the reviewer comments, author rebuttal letters and editorial decision letters if the authors agree. Such peer review material is made available as a supplementary peer review file. Please state in the cover letter 'I wish to participate in transparent peer review' if you want to opt in, or 'I do not wish to participate in transparent peer review' if you don't. Failure to state your preference will result in delays in accepting your manuscript for publication.

ORCID

Sincerely,
Rita

Rita Strack, Ph.D.
Senior Editor
Nature Methods

Reviewer #2 (Remarks to the Author):

The authors have responded well to the previous review and strengthened their manuscript. The addition of the organoid model gives more evidence of the novelty of their work. I only have some minor edits for clarity.

1. The authors stated, "First, imaging is performed using well-corrected immersion optics with a 3D PSF that is as much as 250X smaller by volume based on the measured resolution in either case 38." This comparison is not under the same experimental conditions. For reference 38, they imaged much scattering mice with small NA objectives due to confined working distance and longer wavelengths for in vivo imaging. To demonstrate the resolution superiority of the proposed technique, the authors could consider evaluating the resolution achieved under similar conditions (such as similar NA of objectives, wavelengths and samples).

2. The authors claimed that "Furthermore, the labeling scheme achieves cell-permeability and so one is not limited to intravenous injection as a delivery mechanism."

Is this labeling method applicable to mice? In Reference 38, the authors labeled cells in mice by intravenous injection. Different admission methods can be used for different purposes. Have the authors demonstrated this in vivo? In the revised manuscript, the authors did not show mouse results. If

the authors want to claim this, more results related to in vivo mouse labeling by taking advantage of cell-permeability will be required.

3. In the revised manuscript, the authors claimed the advantages of IR2 over NIR-II imaging. In these claims, the authors have confined the application of IR2 to visible and NIR-I windows, such as “IR2 uses only widespread commercially available dyes and inexpensive silicon-based cameras, rather than custom synthesized dyes and costly InGaAs models.” Therefore, it is necessary to modify the term “infrared” in the title to ensure consistency with the content of the text.

4. About the water absorption, if the authors would like to claim the exact absorption difference between different windows, a measured spectrum by themselves is recommended as the absorption of water is quite different in the literature. The following results are quite different from the references listed in the revised manuscript:

(1) Laura A. Sordillo, Sebastião Pratavieira, Yang Pu, Kaliris Salas-Ramirez, Lingyan Shi, Lin Zhang, Yury Budansky, and R. R. Alfano "Third therapeutic spectral window for deep tissue imaging", Proc. SPIE 8940, Optical Biopsy XII, 89400V (17 March 2014); <https://doi.org/10.1117/12.2040604>

(2) The following result is measured by Jasco equipment.

(<https://jascoinc.com/applications/water-analysis-uv-visible-spectrophotometer/>)

5. The authors used laser powers of 5-20 mW (GFP) and 10-50 mW (CF800). How can the authors prove that these power ranges are not harmful to small animals? Any references or demonstrations?

6. In the newly added Supplementary Fig. 10, why do the GFP data show stronger background in the area without a sample than AF647 data as shown below (The intensity range was adjusted to show the background)? This difference can also be observed in the curves of “Fluorescence intensity” in Supplementary Fig. 10.

7. IR data should be added in Fig. 5 to compare the difference between GFP, IR, and IR2.

8. For Fig. 3, have the authors tried to image fishes older than 4 dpf? Maybe older fish imaging will show apparent differences between GFP and IR results.

9. Providing instructions on how to use the code will enhance its usability and accessibility for users.

Author Rebuttal, first revision:

Response to Reviewer’s Comments:

1. The authors stated, “First, imaging is performed using well-corrected immersion optics with a 3D PSF that is as much as 250× smaller by volume based on the measured resolution in either case 38.” This comparison is not under the same experimental conditions. For reference 38, they imaged much scattering mice with small

NA objectives due to confined working distance and longer wavelengths for in vivo imaging. To demonstrate the resolution superiority of the proposed technique, the authors could consider evaluating the resolution achieved under similar conditions (such as similar NA of objectives, wavelengths and samples).

The reviewer is correct in their observation that the difference in spatial resolution is largely a result of differences in NA and imaging wavelength. However, despite the spatial resolution following well-characterized scaling laws, it becomes more difficult to extract the theoretical performance out of an imaging system at higher NA, where the tolerances to aberrations become smaller. While it is impractical for us to evaluate the microscope under similar conditions, or for the same samples as the previous study used as a point of comparison, we feel the comparison is still worthwhile. We have extended the discussion around this point to highlight the reason for the resolution discrepancy:

“Unlike others who have used convolutional neural networks to restore image quality on the basis of reduced scattering at longer wavelengths⁵⁹, IR² offers the following advantages. First, imaging is performed using well-corrected immersion optics with a 3D PSF that is as much as 250× smaller by volume based on the measured resolution in either case³⁸. While much of this resolution scaling can be understood by comparison of the numerical apertures and imaging wavelengths employed, we note that achieving resolution congruent with the higher NA of our study, requires careful consideration of aberrations.”

Nevertheless, we do note the discrepancy in how the resolution is determined in either case. We have taken the standard of fluorescent microscopies to use sub-diffraction limit sized fluorescent microspheres to measure the PSF, whereas the previous study used a practical definition of resolution in tissue, that being, the smallest vessel that can be resolved. Clearly, this is a different measure of resolution but is no less useful. We

now highlight this fact in the manuscript and additionally have added a Supplementary Note characterizing the spatial resolution in tissue for the IR-mSPIM microscope. We believe this is helpful in characterizing the achievable spatial resolution for the system under weakly (zebrafish) and strongly (drosophila) scattering conditions.

“Nevertheless, a direct comparison is difficult since previously reported NIR-II light sheet systems define the resolution as measured in tissue, which introduces a sample dependency. The resolution in tissue for IR² is explored in Supplementary Note 4.”

2. The authors claimed that “Furthermore, the labeling scheme achieves cell-permeability and so one is not limited to intravenous injection as a delivery mechanism.”

Is this labeling method applicable to mice? In Reference 38, the authors labeled cells in mice by intravenous injection. Different admission methods can be used for different purposes. Have the authors demonstrated this in vivo? In the revised manuscript, the authors did not show mouse results. If the authors want to claim this, more results related to in vivo mouse labeling by taking advantage of cell-permeability will be required.

We thank the reviewer for the comment but believe that there is a misunderstanding in this case. We do not claim that the labelling method is applicable to adult mice as the size of the sample would preclude whole body staining. Given the similar size of embryos, however, we could imagine that the labelling method would work in mouse embryos. Furthermore, we have did not intend to claim that our method works with *in vivo* staining. However, we understand that the original text is misleading when reading only this section:

“Furthermore, the labeling scheme achieves cell-permeability”

The cell-impermeability of dyes in live-cells is a large part of the motivation for the deep learning-based image restoration since if the dyes were cell permeable in vivo, one could in principle use them for in vivo staining using e.g., SNAP/HALO tagging systems. The intention was to highlight that to our knowledge, the various NIR-II dyes have not been used to stain sub-cellular components with molecular precision in whole animals, only in cell culture (e.g., Zhu et al. PNAS, 2017). Whereas in whole animals, mechanical delivery mechanisms such as intravenous delivery have been required. To clarify further we have expanded the statement as such:

“Furthermore, the labeling scheme achieves cell-permeability and complete staining in a few hours in fixed tissues and so samples may be stained via simple immersion rather than requiring intravenous injection as a delivery mechanism.”

3. In the revised manuscript, the authors claimed the advantages of IR2 over NIR-II imaging. In these claims, the authors have confined the application of IR2 to visible and NIR-I windows, such as “IR2 uses only widespread commercially available dyes and inexpensive silicon-based cameras, rather than custom synthesized dyes and costly InGaAs models.” Therefore, it is necessary to modify the term “infrared” in the title to ensure consistency with the content of the text.

The reviewer’s reasoning is fair and in keeping with previous studies performing light sheet imaging in the NIR-II (e.g., Wang et al., Nature Methods, 2020), we have changed the title to reflect that imaging is confined to the near infrared.

4. About the water absorption, if the authors would like to claim the exact absorption difference between different windows, a measured spectrum by themselves is recommended as the absorption of water is quite different in the literature. The following

results are quite different from the references listed in the revised manuscript:

(1) *Laura A. Sordillo, Sebastião Pratavieira, Yang Pu, Kaliris Salas-Ramirez, Lingyan Shi, Lin Zhang, Yury Budansky, and R. R. Alfano "Third therapeutic spectral window for deep tissue imaging", Proc. SPIE 8940, Optical Biopsy XII, 89400V (17 March 2014); <https://doi.org/10.1117/12.2040604>*

(2) *The following result is measured by Jasco equipment.*

(<https://jascoinc.com/applications/water-analysis-uv-visible-spectrophotometer/>)

We thank the reviewer for the comment. We had expanded the discussion around specific water absorption windows as the reviewer had questioned the approximate value provided for water absorption scaling across the NIR-I to the longwave end of the NIR-II. The reference included by the reviewer shows a smaller increase in absorption from water. However, the axis of the respective graph is not labelled, so one cannot infer absolute, but rather only relative, absorption values. Furthermore, the link provided to the Jasco spectrophotometer only shows wavelengths to 850 nm. However, all three studies that were originally cited agree within a small margin of error (typically < 10% over a range of > 4 orders of magnitude).; the literature is full of further studies covering sections of the spectral range (e.g. D. M. Wieliczka and S. Weng and M. R. Querry, "Wedge shaped cell for highly absorbent liquids: infrared optical constants of water," *Appl. Opt.*, 28, 1714-1719, (1989), R. C. Smith and K. S. Baker, "Optical properties of the clearest natural waters (200 - 800nm)," *Appl. Opt.*, 20, 177--184, (1981), V. M. Zolotarev, B. A. Mikhailov, L. L. Alperovich, S. I. Popov, "Dispersion and absorption of liquid water in the infrared and radio regions of the spectrum," *Optics and Spectroscopy*, 27, 430-432 (1969) and which all agree to a considerable degree over their overlapping spectral ranges with the more complete data included in the paper. Since there are so many studies in agreement in this regard we do not intend to make measurements of water absorption. We do, however, believe that the discussion goes far beyond that

which is warranted for this paper but that the increased absorption is worth noting in the context of NIR imaging as such we have shortened the discussion as given below:

“Although a light sheet microscope compatible with dyes emitting up to 1700 nm has been reported³⁸, absorption from water increases substantially from the visible to NIR-I and from NIR-I to NIR-II. While absorption is already appreciable in the NIR-I, this range is commonly considered an optimal window where scattering and autofluorescence are strongly suppressed relative to the blue-green while the increased absorption does not cause major heating of tissue or attenuation of the excitation/emission light.”

We feel that the amended text places enough emphasis on the absorption from visible to NIR-I/NIR-II to be helpful to readers without making quantitative claims, which in any case only reflect on absorption by water without considering other tissue components.

5. The authors used laser powers of 5-20 mW (GFP) and 10-50 mW (CF800). How can the authors prove that these power ranges are not harmful to small animals? Any references or demonstrations?

We appreciate the opportunity to demonstrate that our technique is minimally harmful to small animals as this is one of the hallmarks of light sheet-based imaging. Since CF800 is used only for fixed tissue imaging to provide the training data, we believe this aspect of the question is not relevant. However, the potentially deleterious nature of the GFP imaging (488 nm excitation, 5 – 20 mW) in live animals is certainly relevant. For the laser power data, we have provided the power at the laser source assuming that the manufacturer stated maximum power corresponds to the maximum output power of the laser engine/laser head, thus representing the worst possible scenario for rates of photodamage. However, to make comparison with other literature sources, it is necessary to report laser power used at the sample itself. We have measured the associated light throughput efficiency, which is on the order of 10% (given a number of

slits/lenses and mirrors, each of which attenuate the beam). We have adjusted the discussion in the manuscript accordingly:

“For the zebrafish samples, stacks were generated acquiring a Z plane every 5 μm using laser powers of < 2.2 mW (GFP) and <3.4 mW (CF800) and exposure times of 100 ms for both. Note, laser powers are given as measured in sample media downstream of the illumination objective. For the Drosophila samples, volume stacks were generated acquiring a Z plane every 2.5 μm using laser powers of < 1.1 mW (GFP) and < 3.4 mW (CF800) and exposure times of 100 ms for both.”

To explore whether the laser powers used can indeed be considered as safe for small animal imaging, we have added Supplementary Note 5:

“Live zebrafish and Drosophila samples were imaged using laser powers in the previously stated range for fixed tissue GFP imaging. However, the exposure time was set to 20 ms, which is more typical for light sheet based live imaging. The laser powers used are comparable to those of previous studies of unimpeded biological development and function using light sheet microscopy (Supplementary Note 5).”

“Supplementary Note 5. Comparison of total effective laser exposure used for live imaging with light sheet fluorescence microscopy.

To explore whether the live imaging reported herein is performed within safe ranges of laser exposure for delicate developing embryos/larvae we must first consider the laser power at the sample. We used < 2.2 mW, <1.1 mW laser power at 488 nm for live imaging of zebrafish and drosophila larva/embryos respectively. For approximately equivalent light-sheet imaging schemes the photon burden experienced by the live sample can be described by the laser power summed for the total number of exposures

over the duration of the experiment. For the zebrafish (ca. 72 hrs, dt = 300 s, one color channel) and drosophila (ca. 24 hrs, dt = 300 s, one color channel) data presented, we calculate 1,848 and 275 mW equivalent exposures respectively. For comparison Schmid et al. used 17,280 mW equivalent exposures for early imaging of zebrafish embryogenesis (6 mW, ca. 12 hrs, dt = 30 s, one color channel) (ref Schmid et al. Nature Comms, 2013). Shah et al. used 13,824 mW for conceptually similar imaging (8 mW, ca. 12 hrs, dt = 150 s, three color channels) (ref: Shah et al. Nature Communications, 2019). Weber et al. used 2 mW at the illumination objective back aperture (ca. 1.6 mW at the object assuming 80% transmission through the objective lens), which was lower than the 5 mW (4 mW at the object) threshold for a measurable increase in heart rate in the zebrafish embryo/larva. (ref: Weber et al. eLife, 2017). Chhetri et al. performed developmental imaging of drosophila using 2,160 mW equivalent exposures (0.1 mW, ca. 3 hrs, dt = 4 s, two color channels).. In this case since the light sheet was produced temporally by line scanning, the instantaneous or peak intensity is far higher than the other examples, or indeed the studies presented herein, making direct comparison difficult. (Chhetri et al., Nature Methods, 2015). Despite the challenges in directly comparing the different studies, we note that the <2.1 mW laser power used at 488 nm for zebrafish is below the threshold shown by Weber et al. for perturbation of the heart beat period. The mW equivalent exposures being substantially lower than the other examples considered demonstrates that the photon burden associated with live imaging presented is well within a normal range for live imaging of developing embryos.”

We further note that we now have > 20 years of experience in developing and using light sheet microscopes with developing embryos; the success of these studies and the fact that light sheet microscopy represents the gold standard in long-term gentle time-

lapse fluorescence imaging provides sufficient justification that the laser powers presented are safe for small animal imaging.

6. In the newly added Supplementary Fig. 10, why do the GFP data show stronger background in the area without a sample than AF647 data as shown below (The intensity range was adjusted to show the background)? This difference can also be observed in the curves of “Fluorescence intensity” in Supplementary Fig. 10.

The images in Supplementary Figure 10 are shown with the automated brightness and contrast adjustment of the image visualization package used. On the other hand, to provide a fairer comparison, the fluorescence intensity profiles are normalized to the 3rd and 99.7th percentiles of each line separately. The background visible in the deeper GFP images is likely due to the out-of-focus GFP fluorescence coming from the sample, which is situated between the focal plane and the imaging objective. To provide a better visualization of the different planes, we have now normalized the images using their 3rd and 99.7th percentiles.

7. IR data should be added in Fig. 5 to compare the difference between GFP, IR, and IR2.

As the reviewer correctly noticed, Fig. 5 does not show IR images of the pescoids data. This is because Fig. 5 contains images of live pescoids, which have not been stained with IR dyes. Instead, we included in Supplementary Fig. 10 the GFP and IR images of fix-and-stained pescoids. Following the reviewer comment, we will add restored IR2 images for the fixed-and-stained pescoids shown in Supplementary Fig. 10.

8. For Fig. 3, have the authors tried to image fishes older than 4 dpf? Maybe older fish imaging will show apparent differences between GFP and IR results.

We thank the reviewer for the comment regarding imaging in older fish. The conjecture that older fish may show more benefit due to their larger size may prove to be correct. Unfortunately, our animal protocol does not allow for experiments in older fish, which also require food to continue to grow. Moreover, older fish are also not as interesting from a developmental biology perspective since the basic body plan including all major organs has formed after 4 days.

9. Providing instructions on how to use the code will enhance its usability and accessibility for users.

A github repository with the code used for training and restoration of raw data, as well as the code used to generate all figures in the manuscript, exists:

https://github.com/grinic/2023_InfraRed_Image_Restoration.

We thank the reviewer's comment, and we have now implemented a README file to assist users interested in applying our approach to their own datasets.

Final Decision Letter:

Dear Jan,

I am pleased to inform you that your Article, "Image restoration of degraded time-lapse microscopy data mediated by near-infrared imaging.", has now been accepted for publication in Nature Methods. Your paper is tentatively scheduled for publication in our February print issue, and will be published online prior to that. The received and accepted dates will be Nov 10, 2022 and Nov 10, 2023. This note is intended to let you know what to expect from us over the next month or so, and to let you know where to address any further questions.

Over the next few weeks, your paper will be copyedited to ensure that it conforms to Nature Methods style. Once your paper is typeset, you will receive an email with a link to choose the appropriate publishing options for your paper and our Author Services team will be in touch regarding any additional information that may be required.

You will receive a link to your electronic proof via email with a request to make any corrections within 48 hours. If, when you receive your proof, you cannot meet this deadline, please inform us at rjsproduction@springernature.com immediately.

Please note that *Nature Methods* is a Transformative Journal (TJ). Authors may publish their research with us through the traditional subscription access route or make their paper immediately open access through payment of an article-processing charge (APC). Authors will not be required to make a final decision about access to their article until it has been accepted. [Find out more about Transformative Journals](https://www.springernature.com/gp/open-research/transformative-journals)

Your paper will now be copyedited to ensure that it conforms to Nature Methods style. Once proofs are generated, they will be sent to you electronically and you will be asked to send a corrected version within 24 hours. It is extremely important that you let us know now whether you will be difficult to contact over the next month. If this is the case, we ask that you send us the contact information (email, phone and fax) of someone who will be able to check the proofs and deal with any last-minute problems.

If, when you receive your proof, you cannot meet the deadline, please inform us at rjsproduction@springernature.com immediately.

Once your manuscript is typeset and you have completed the appropriate grant of rights, you will receive a link to your electronic proof via email with a request to make any corrections within 48 hours. If, when you receive your proof, you cannot meet this deadline, please inform us at rjsproduction@springernature.com immediately.

Once your paper has been scheduled for online publication, the Nature press office will be in touch to confirm the details.

Once your paper has been scheduled for online publication, the Nature press office will be in touch to confirm the details.

Content is published online weekly on Mondays and Thursdays, and the embargo is set at 16:00 London time (GMT)/11:00 am US Eastern time (EST) on the day of publication. If you need to know the exact publication date or when the news embargo will be lifted, please contact our press office after you have submitted your proof corrections. Now is the time to inform your Public Relations or Press Office about your paper, as they might be interested in promoting its publication. This will allow them time to prepare an accurate and satisfactory press release. Include your manuscript tracking number NMETH-A50903B and the name of the journal, which they will need when they contact our office.

About one week before your paper is published online, we shall be distributing a press release to news organizations worldwide, which may include details of your work. We are happy for your institution or funding agency to prepare its own press release, but it must mention the embargo date and Nature Methods. Our Press Office will contact you closer to the time of publication, but if you or your Press Office have any inquiries in the meantime, please contact press@nature.com.

Nature Portfolio journals [encourage authors to share their step-by-step experimental protocols](https://www.nature.com/nature-research/editorial-policies/reporting-standards#protocols) on a protocol sharing platform of their choice. Nature Portfolio 's Protocol

Exchange is a free-to-use and open resource for protocols; protocols deposited in Protocol Exchange are citable and can be linked from the published article. More details can be found at www.nature.com/protocolexchange/about.

Best regards,
Rita

Rita Strack, Ph.D.
Senior Editor
Nature Methods